# RQ-MoE: Residual Quantization via Mixture of Experts for Efficient Input-Dependent Vector Compression

**Zhengjia Zhong**[1]  **Shuyan Ke**[1]  **Zaizhou Lin**[1]  **Jiaqi Song**[1]  **Hongyi Lan**[1]  **Hui Li**[1]

## Abstract

Vector quantization is a fundamental tool for compressing high-dimensional embeddings, yet existing multi-codebook methods rely on static codebooks that limit expressiveness under heterogeneous data geometry. While recent dynamic quantizers like QINCo adapt codebooks to individual inputs and improve expressiveness, their strict sequential dependencies create decoding bottlenecks. We propose Residual Quantization via Mixture of Experts (RQ-MoE), a framework combining a two-level MoE with dual-stream quantization to enable input-dependent codebook adaptation for efficient vector quantization. RQ-MoE enables dynamic codebook construction and decouples instruction from quantization, facilitating parallel decoding. Theoretically, we show that standard Residual Quantization and QINCo can be recovered as constrained special cases of RQ-MoE, and derive a guideline for setting expert dimensionality in RQ-MoE. Extensive experiments show that RQ-MoE achieves state-of-the-art or on-par performance in reconstruction and retrieval, while it can provide 6×–14× faster decoding than prior vector quantization methods. The implementation is available at https://github.com/KDEGroup/RQ-MoE.

## 1. Introduction

Vector embeddings have become the fundamental representation paradigm for high-dimensional data in modern machine learning. Increasing embedding dimensionality often enhances representation expressiveness, but at the cost of increased difficulty in manipulating and understanding

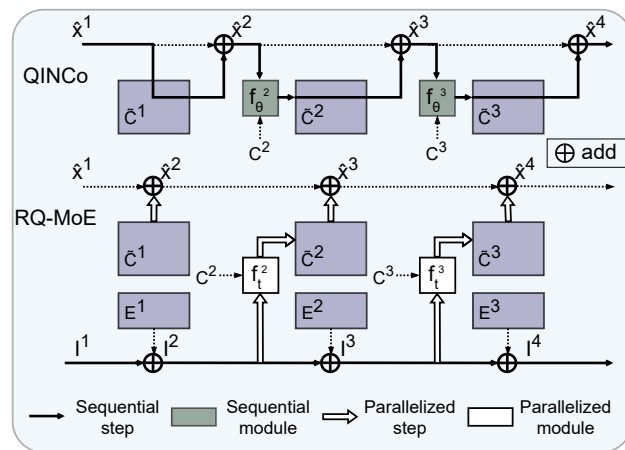

*Figure 1.* Decoding comparison of QINCo and RQ-MoE. QINCo is limited by strict serial dependencies in both decoding steps and its $f_\theta$ component, while RQ-MoE achieves inter-step parallelism via a fast path and intra-step parallelism through $f_t$ (Eq. 6).

(the high-dimensional curse, Donoho, 2000), substantial storage and computational overhead, especially in large-scale deployment (Kaplan et al., 2020; Zhang et al., 2023). This tension makes vector compression a critical component of modern systems, as it enables compact representations that significantly reduce memory footprint and inference cost while retaining the essential information required for accurate downstream tasks (May et al., 2019).

Vector Quantization (VQ; Gray, 1984) is a representative compression paradigm that maps each vector to a "prototype" vector via learning a codebook of centroids from the training vectors. However, to achieve low quantization error, the codebook size must grow exponentially with the target bit rate. To remedy this issue, multi-codebook quantization (MCQ) methods assign each vector with elements from multiple codebooks. Residual Quantization (RQ; Chen et al., 2010) is one exemplifying MCQ approach that adopts a recursive coarse-to-fine approximation strategy, representing a vector with multiple codewords selected sequentially from a series of smaller codebooks. RQ has shown its power in recommender systems and beyond (Lee et al., 2022; Zeghidour et al., 2022; Rajput et al., 2023).

Prior VQ methods, including RQ-based approaches, rely on

---

[1]Key Laboratory of Multimedia Trusted Perception and Efficient Computing, Ministry of Education of China, Xiamen University, Xiamen, China. Correspondence to: Hui Li <hui@xmu.edu.cn>.

*Proceedings of the 43rd International Conference on Machine Learning*, Seoul, South Korea. PMLR 306, 2026. Copyright 2026 by the author(s).

static codebooks, limiting expressiveness under heterogeneous data geometry. From a manifold learning perspective, vector compressibility arises from the fact that data distributions concentrate on low-dimensional manifolds (Tenenbaum et al., 2000; Roweis & Saul, 2000). Using static codebooks implicitly assumes homogeneous local geometry across different regions of the embedding space, an assumption that rarely holds in complex real-world datasets. Recent dynamic-codebook quantizers QINCo (Huijben et al., 2024) and its successor (Vallaeys et al., 2026) adapt codebooks at each quantization step of RQ. But they introduce strict sequential dependencies that substantially hinder decoding efficiency as demonstrated in Fig. 1.

The fundamental limitation of static codebooks lies in the rigid partitioning of the vector space. Since different regions of the embedding manifold may require different quantization strategies, it is desirable to employ *conditional computation*, where the quantizer activates specialized codebooks tailored to the vector's local structure. Along this direction, Mixture of Experts (MoE, Jacobs et al., 1991) is a natural solution as it scales capacity through dynamically activating expert sub-networks based on the local characteristics of the input distribution, without a proportional increase in inference cost. Nevertheless, directly applying MoE to quantization requires storing expert routing decisions for each input, which increases bit overhead and complicates optimization (see discussion in Sec. 4.1).

To alleviate the issues of using static codebooks while avoiding increasing overhead, we propose Residual Quantization via Mixture of Experts (RQ-MoE), a novel framework that leverages a two-level MoE with a dual-stream quantization process to enable input-dependent codebook adaptation for efficient vector quantization. At its core, the two-level MoE mechanism enables the dynamic construction of input-dependent codebooks, while the dual-stream quantization decouples instruction from quantization, allowing specialized codebooks to be instantiated independently and supporting parallel reconstruction.

The contributions of this work are summarized as follows:

- **Input-Dependent, Index-Reused RQ-MoE Framework:** We introduce the two-level MoE residual quantization framework that achieves input-dependent codebook adaptation with zero additional bit overhead. Routing is embedded in the quantization indices themselves, so each index simultaneously determines the expert selection for the next step.

- **Efficiency Boosting with Parallelizable Decoding:** We propose dual-stream quantization that decouples the instruction and reconstruction paths, removing the sequential dependency present in prior dynamic quantizers and enabling fully parallel decoding.

- **Reconstruction with Normalized Residual Loss:** To enhance robust training, we design a novel Normalized Residual Loss (NRL) for reconstruction. By normalizing the contribution of each step based on the ratio of successive residuals, NRL stabilizes the training of deep-level experts and reduces sensitivity to outliers.

- **Theoretical Analysis:** We prove that standard RQ and the previous dynamic-codebook quantization method QINCo can be recovered as constrained special cases of RQ-MoE, and derive a guideline for setting expert dimensionality in RQ-MoE. We further analyze the complexity of RQ-MoE and existing methods, demonstrating the efficiency of RQ-MoE.

Experiments show that RQ-MoE achieves state-of-the-art or on-par performance in reconstruction and retrieval, while it can provide $6\times$–$14\times$ faster decoding than prior methods.

## 2. Related Work

**Static-Codebook Vector Quantization.** The classical method VQ (Gray, 1984) maps each vector to a single codeword selected from a codebook of size $K$, resulting in limited representational capacity and high reconstruction distortion. Multi-codebook Quantization (MCQ) addresses this limitation by employing multiple codebooks. Various MCQ methods (Ge et al., 2013; Babenko & Lempitsky, 2014; Martinez et al., 2018) have been proposed. PQ (Jégou et al., 2011a) partitions vectors into independent subspaces and quantizes each subspace using a low-dimensional codebook, reconstructing vectors through a Cartesian product. RQ (Chen et al., 2010) adopts a coarse-to-fine strategy by successively quantizing the residuals from previous steps. More recent neural MCQ methods, such as UNQ (Morozov & Babenko, 2019), leverage deep networks to determine codeword indices from learned representations rather than raw Euclidean distances. Compared to RQ-MoE, these approaches rely on static codebooks at inference, where the quantization space remains fixed and fails to adapt to vector diversity.

**Dynamic-Codebook Vector Quantization.** QINCo (Huijben et al., 2024) raises the concept of data-dependent codebooks by generating specialized codebooks at each step, conditioning a residual-MLP on the partial reconstruction from previous steps. This design significantly improves reconstruction and retrieval accuracy over static baselines. However, it introduces a strong sequential dependency, as each step relies on the outputs of its predecessors. While its recent successor QINCo2 (Vallaeys et al., 2026) improves efficiency through approximate predictors and candidate pre-selection, it further relies on beam search to compensate for the resulting accuracy degradation, leaving the decoding process inherently sequential.

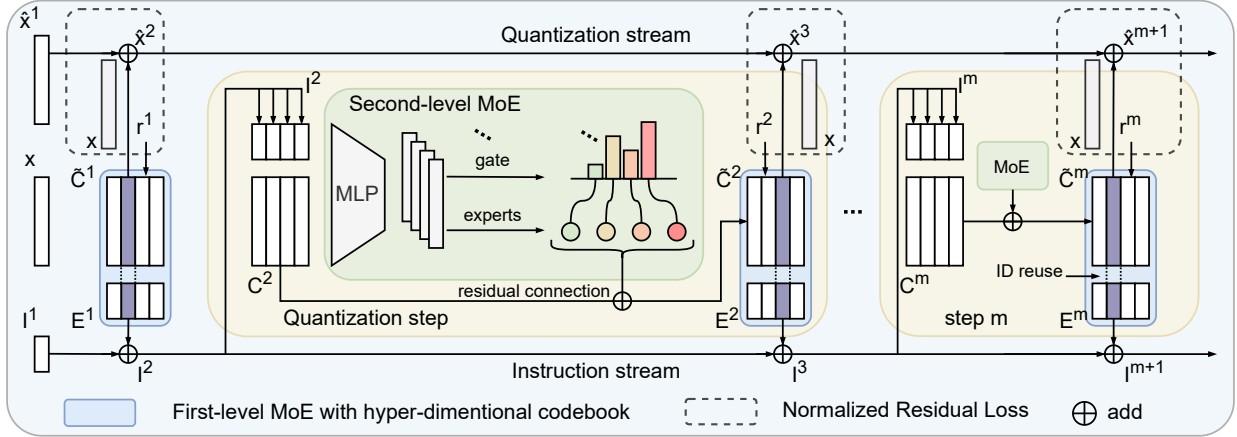

*Figure 2.* Overview of RQ-MoE. It features a dual-stream design equipped with a two-level MoE mechanism. The Instruction Stream (bottom) leverages hyper-dimensional codebooks and first-level MoE to accumulate expert signals via implicit routing. These signals condition the Quantization Stream (top), where the second-level MoE dynamically adapts base codebooks for precise residual reconstruction. This decoupled structure supports efficient parallel decoding.

**Mixture of Experts.** Mixture of Experts (MoE) (Jacobs et al., 1991) mitigates parameter interference and crosstalk in large-scale neural networks by dynamically routing inputs to a set of parallel expert sub-networks. MoE significantly expands model capacity and expressive power while keeping inference cost nearly constant (Lepikhin et al., 2021; Jiang et al., 2024). Hence, it has become the prevalent architecture for constructing Large Language Models (Cai et al., 2025). Modern implementations often employ sparse gating to activate only a subset of experts (Shazeer et al., 2017), improving computational efficiency and stability (Fedus et al., 2022; DeepSeek-AI, 2024).

## 3. Preliminary

**Residual Quantization.** RQ maps a high-dimensional vector $\mathbf{x} \in \mathbb{R}^D$ into a combination of discrete indices. Formally, we define a set of $M$ codebooks indexed by $m \in \{1, \ldots, M\}$, where each codebook $\mathcal{C}^m = \{\mathbf{c}_k^m\}_{k=1}^K \subset \mathbb{R}^D$ contains $K$ centroids (i.e., codewords) of dimension $D$. The quantization process proceeds iteratively through $M$ stages:

1. **Initialization:** Let the initial reconstruction vector be $\hat{\mathbf{x}}^1 = \mathbf{0}$, and the initial residual be $\mathbf{r}^1 = \mathbf{x}$.

2. **Recursive Approximation:** At each stage $m$, RQ performs the nearest neighbor search to select the optimal index $i^m$ from the current codebook $\mathcal{C}^m$:

$$i^m = \arg \min_{k \in \{1, \ldots, K\}} \|\mathbf{r}^m - \mathbf{c}_k^m\|_2^2. \tag{1}$$

3. **Residual Update:** The reconstruction vector is updated as $\hat{\mathbf{x}}^{m+1} = \hat{\mathbf{x}}^m + \mathbf{c}_{i^m}^m$, and the residual for the subsequent stage is computed as $\mathbf{r}^{m+1} = \mathbf{x} - \hat{\mathbf{x}}^{m+1}$.

Ultimately, the original vector $\mathbf{x}$ is represented by the sequence of indices $(i^1, i^2, \ldots, i^M)$. The final reconstructed vector $\hat{\mathbf{x}}$ is obtained by the summation of the selected codewords across all stages: $\hat{\mathbf{x}} = \sum_{m=1}^M \mathbf{c}_{i^m}^m$, and RQ is optimized by minimizing the reconstruction loss between residual values and $\mathbf{x}$.

**Mixture of Experts.** MoE scales model capacity by conditioned activation of sub-networks. Given $N$ expert functions $\{\mathcal{E}_i\}_{i=1}^N$ and a gating network (router) $G(\cdot)$, it defines the output $\mathbf{y}$ for an input $\mathbf{v}$ as:

$$\mathbf{y} = \sum_{i \in \mathcal{T}} G(\mathbf{v})_i \mathcal{E}_i(\mathbf{v}), \tag{2}$$

where $\mathcal{T}$ are indices selected by the router (typically the top-$Q$ entries) and $Q \ll N$, ensuring that only a fraction of parameters are activated per input.

## 4. Methodology

Fig. 2 depicts RQ-MoE. At a high level, it combines a two-level MoE mechanism with a dual-stream quantization process to achieve adaptive codebook selection and parallelizable decoding without increasing overhead. The first-level MoE determines *where* a vector lies in the discrete representation space, while the second-level MoE determines *how* that region is locally approximated. The dual-stream quantization decouples the instruction stream from the quantization stream, allowing specialized codebooks to be instantiated independently and supporting parallel reconstruction.

Specifically, RQ-MoE conducts implicit routing with hyper-dimensional codebooks and index reuse in the first-level MoE (Sec. 4.1). In dual-stream quantization, the *instruction stream* propagates expert signals across quantization

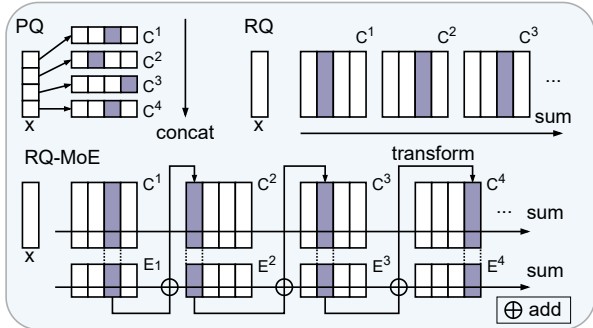

*Figure 3.* Comparison of quantization architectures. PQ uses sub-dimensional codebooks; RQ employs codebooks of the same dimension as the input; RQ-MoE utilizes an hyper-dimensional codebook to capture local manifold information.

steps, while the *quantization stream* performs residual quantization using input-adaptive codebooks operated via the second-level MoE (Sec. 4.2). To facilitate robust training, we further design a Normalized Residual Loss for guiding reconstruction (Sec. 4.3).

Together, these components enable high-fidelity reconstruction, efficient parallel inference, and effective utilization of input-dependent local manifold structure.

### 4.1. Implicit Routing with Hyper-Dimensional Codebook and Index Reuse

A straightforward adaptation of MoE to RQ might involve multiple codebooks and an explicit gating network. However, this naive integration has two issues: (i) storing expert routing decisions requires additional bits beyond the quantization indices (e.g., selecting among 4 experts requires 2 extra bits, resulting in a 25% storage overhead for a 256-entry codebook), and (ii) under a fixed bit budget, allocating bits to explicit expert selection is inherently inefficient. It is often more effective to utilize this capacity to expand the codebook size directly, rather than consuming the budget on routing overhead.

From a functional perspective, RQ can be interpreted as a degenerate MoE process, where nearest-neighbor assignment acts as implicit routing and the selected codeword serves as a top-1 expert.

To better leverage the power of MoE, RQ-MoE fuses expert routing with quantization through *hyper-dimensional codebooks* and *index reuse* in the first-level MoE. Instead of relying on a separate gating network, the same codeword selection operation simultaneously determines the quantized output and the activated expert component.

As illustrated in Fig. 3, we define a codebook $\mathcal{W}^m = \{\mathbf{w}_k^m\}_{k=1}^K$ for each quantization step $m$, where each entry $\mathbf{w}_k^m \in \mathbb{R}^{D+D_e}$ is a $(D+D_e)$-dimensional vector, which

can be partitioned into two functional components:

$$\mathbf{w}_k^m = [\mathbf{c}_k^m; \mathbf{e}_k^m], \qquad (3)$$

where $\mathbf{c}_k^m \in \mathbb{R}^D$ denotes the quantization component used for residual matching, and $\mathbf{e}_k^m \in \mathbb{R}^{D_e}$ represents the expert component that encodes local manifold characteristics of the input. For notational convenience, we refer to the collection $\{\mathbf{c}_k^m\}_{k=1}^K$ as the *base codebook* $\mathbf{C}^m$ and $\{\mathbf{e}_k^m\}_{k=1}^K$ as the *expert codebook* $\mathbf{E}^m$, while both components share the same index throughout the quantization process.

The index $i^m$ is obtained via Eq. (1), and it also acts as the router $G(\cdot)$ in the first-level MoE. Once $i^m$ is selected, the corresponding expert vector $\mathbf{e}_{i^m}^m$ is activated and passed as the instruction to the next step, ensuring that routing decisions are directly aligned with the quantization trajectory.

### 4.2. Dual-Stream Quantization

Existing dynamic quantizers tightly couple conditioning generation with reconstruction, resulting in entangled optimization dynamics and inherently sequential decoding dependencies. RQ-MoE removes this coupling by decoupling instruction generation from reconstruction, enabling both parallel decoding and more flexible contextual adaptation.

The core architecture of RQ-MoE follows the iterative refinement logic of RQ and operates within a dual-stream framework consisting of an instruction stream and a quantization stream. The second-level MoE uses the instruction stream to transform the base codebooks into input-specific codebooks at each step, enabling the quantization stream to select codewords that are aligned with the local manifold of the input vector.

The instruction stream maintains a dynamic instruction vector $\mathbf{I}^m \in \mathbb{R}^{D_e}$, which serves as a latent memory recording the cumulative expert information across quantization steps. At the first step ($m = 1$), the instruction vector is initialized as a zero vector, $\mathbf{I}^1 = \mathbf{0}$. For subsequent steps ($m > 1$), $\mathbf{I}^m$ is updated based on the previous instruction and the selected expert component:

$$\mathbf{I}^m = f_{ins}(\mathbf{I}^{m-1}, \mathbf{E}_{i^{m-1}}^{m-1}) = \mathbf{I}^{m-1} + \mathbf{E}_{i^{m-1}}^{m-1}, \qquad (4)$$

where the additive update is a simple choice to efficiently propagate expert signals and enable parallelizable decoding. This mechanism ensures that the instruction signal $\mathbf{I}^m$ carries the full context of the preceding selections to inform the next quantization step.

The quantization stream operates similarly to RQ, where the input vector is approximated by a sequence of discrete codewords. However, instead of utilizing static base codebooks, it employs *dynamic codebooks* $\tilde{\mathcal{C}}^m$ that are conditioned on the accumulated context $\mathbf{I}^m$ of the instruction stream. For

the $m$-th step, the optimal index $i^m$ is selected by minimizing the Euclidean distance between the current residual $\mathbf{r}^m$ and the dynamic codewords:

$$i^m = \arg \min_{k \in \{1,...,K\}} \|\mathbf{r}^m - \tilde{\mathbf{c}}_k^m\|_2^2, \qquad (5)$$

where $\tilde{\mathbf{c}}_k^m \in \mathbb{R}^D$ denotes the $k$-th codeword of the dynamic codebook at step $m$. The transformation from a static base codebook $\mathcal{C}^m$ to a dynamic codebook $\tilde{\mathcal{C}}^m$ is governed by a transformation function $f_t$, conditioned on the previous instruction $\mathbf{I}^m$:

$$\tilde{\mathbf{c}}_k^m = f_t(\mathbf{c}_k^m, \mathbf{I}^m), \qquad (6)$$

where $f_t$ is implemented as a weighted MoE module. Specifically, each base codeword $\mathbf{c}_k^m$ is concatenated with the instruction vector $\mathbf{I}^m$ and projected via a linear layer to a $D$-dimensional space:

$$\mathbf{z}_k^m = \text{Linear}([\mathbf{c}_k^m; \mathbf{I}^m]). \qquad (7)$$

The projected vector $\mathbf{z}_k^m \in \mathbb{R}^D$ is processed by the second-level MoE comprising a gating network and $N$ experts. The gating network generates mixture weights over the experts via a linear transformation followed by a softmax:

$$\boldsymbol{\alpha}_k^m = \text{softmax}\big(\text{Linear}(\mathbf{z}_k^m)\big), \qquad (8)$$

where $\boldsymbol{\alpha}_k^m \in \mathbb{R}^N$ denotes the weights for the $N$ experts.

Each expert $\mathcal{E}_n^m$ is implemented as an $L$-layer MLP, where each layer follows a bottleneck residual structure with an internal expansion to $H$ dimensions and ReLU activation:

$$\mathbf{h}_l = \mathbf{h}_{l-1} + \text{MLP}_l(\mathbf{h}_{l-1}), \quad l = 1, \dots, L, \qquad (9)$$

and the input-output relation for the expert is

$$\mathcal{E}_n(\mathbf{z}_k^m) = \mathbf{h}_L, \quad \mathbf{h}_0 = \mathbf{z}_k^m. \qquad (10)$$

Expert outputs are aggregated according to gating weights:

$$\Delta\mathbf{c}_k^m = \sum_{n=1}^{N} \boldsymbol{\alpha}_{k,n}^m \, \mathcal{E}_n(\mathbf{z}_k^m), \qquad (11)$$

and the updated codeword is obtained by adding the deformation to the base codeword:

$$\tilde{\mathbf{c}}_k^m = \mathbf{c}_k^m + \Delta\mathbf{c}_k^m. \qquad (12)$$

For the first quantization step, the base codebook is used directly as the dynamic codebook, i.e., $\tilde{\mathcal{C}}^1 = \mathcal{C}^1$. By combining multiple expert deformations, the resulting dynamic codewords better align with local structure of the input's manifold. This enables the instruction stream to guide input-dependent adaptations in subsequent steps, effectively tailoring each codebook to the local geometry of the data.

The two-level MoE performs a coarse-to-fine adaptation process, where the first-level MoE accumulates contextual routing information through the instruction stream, while the second-level MoE dynamically combines expert basis functions to adapt the codebook within localized subspaces.

## 4.3. Reconstruction with Normalized Residual Loss

Previous RQ-based methods commonly use either the final-step mean squared error (MSE), which evaluates only the last residual, or a per-step MSE, which supervises all intermediate residuals, as the reconstruction loss. We empirically find that using only the final-step loss leads to unstable training as mentioned by Huijben et al. (2024), since shallow layers receive insufficient gradient signals, whereas applying a per-step MSE tends to overemphasize early residuals at the expense of later steps.

To remedy these problems, we propose the *normalized residual loss* (NRL). NRL measures the contribution of each residual step relative to its predecessor, ensuring that each step is optimized proportionally to its current reconstruction difficulty. Meanwhile, it automatically down-weights extreme residuals, addressing key stability and robustness limitations of the MSE loss.

We first define the relative residual ratio as:

$$\rho^m = \frac{\|\mathbf{r}^{m+1}\|_2^2}{\text{sg}(\|\mathbf{r}^m\|_2^2) + \epsilon}, \qquad (13)$$

where $\text{sg}(\cdot)$ denotes the stop-gradient operator and $\epsilon$ is a small constant for numerical stability. The normalization term $\rho^m$ captures the relative improvement achieved at step $m$, quantifying how much the current step reduces the remaining reconstruction error compared to its predecessor. This sequential formulation naturally aligns with RQ, where later steps are explicitly designed to refine earlier approximations rather than solve parallel objectives. To mitigate instability caused by varying residual scales across steps, we adopt a logarithmic transformation for NRL:

$$\mathcal{L}_{\text{NRL}} = \sum_{m=1}^{M} \log\left(1 + \rho^m\right). \qquad (14)$$

Unlike ratio-based normalization losses commonly used in multi-task learning, where ratios balance gradients across independent objectives (Chen et al., 2018; Kendall et al., 2018), NRL operates along a sequential refinement chain. Each residual $\mathbf{r}^{m+1}$ is not an independent target, but the direct outcome of all preceding quantization steps.

**Comparison to MSE Loss.** NRL not only balances residual contributions across quantization steps but also improves robustness compared to the standard MSE loss.

For a residual $\mathbf{r}^{m+1}$ at quantization step $m$, the gradient of the MSE loss is:

$$\nabla_{\mathbf{r}^{m+1}} \mathcal{L}_{\text{MSE}} = 2\|\mathbf{r}^{m+1}\|_2, \qquad (15)$$

which directly scales with the residual magnitude. As RQ proceeds, residuals typically decrease across steps, causing

gradients for later steps to become progressively smaller and leaving later refinements under-optimized, despite the model's capacity to further reduce reconstruction error. Moreover, the MSE loss exhibits an *unbounded influence function*. When $\mathbf{r}^{m+1}$ is unusually large, the gradient grows proportionally, encouraging overfitting to outliers and potentially leading to gradient explosion.

From a robust estimation perspective, the gradient induced by NRL exhibits the behavior of a *redescending influence function*. In robust statistics, such influence functions are characteristic of redescending M-estimators, which limit the impact of extreme residuals. Specifically, the gradient of NRL with respect to the residual is:

$$\nabla_{\mathbf{r}^{m+1}} \mathcal{L}_{\text{NRL}} = \frac{2\|\mathbf{r}^{m+1}\|_2}{\|\mathbf{r}^m\|_2^2 + \|\mathbf{r}^{m+1}\|_2^2 + \epsilon} = \frac{2\|\mathbf{r}^{m+1}\|_2}{\|\mathbf{r}^{m+1}\|_2^2 + C}, \quad (16)$$

where $\|\mathbf{r}^m\|_2$ is treated as a constant due to the stop-gradient operation. This formulation dynamically normalizes the gradient using both the previous-step residual and the current residual magnitude, preventing early steps from dominating optimization and enabling meaningful updates at later steps. For moderate residual values, the gradient magnitude increases with $\|\mathbf{r}^{m+1}\|_2$, preserving sensitivity to reconstruction errors. However, as $\|\mathbf{r}^{m+1}\|_2 \to \infty$, the gradient gradually approaches zero, yielding a bounded and redescending influence function.

### 4.4. Parallel Inference

Compared to existing dynamic-codebook quantization methods like QINCo (Huijben et al., 2024; Vallaeys et al., 2026), RQ-MoE enables a substantial improvement in inference efficiency, particularly during decoding.

During encoding, RQ-MoE follows the sequential refinement process of RQ. To determine the index sequence $\{i^1, i^2, \ldots, i^M\}$, the residual at each step must be computed to perform the nearest neighbor search in the step-specific codebook $\tilde{\mathcal{C}}^m$. Therefore, the encoding process maintains a sequential dependency similar to existing adaptive neural quantizers. However, in our design, the efficiency of encoding can also be boosted in theory by using more experts.

RQ-MoE brings a substantial efficiency gain during decoding, where the goal is to reconstruct $\hat{\mathbf{x}}$ from a given index sequence. In QINCo, decoding is inherently sequential, as generating the $m$-th dynamic codebook requires access to the cumulative reconstruction from all previous steps $\{1, \ldots, m-1\}$. Even in its subsequent extension (Vallaeys et al., 2026), decoding speedups are primarily achieved via Pairwise Additive Decoder (PAD), which uses larger codebooks for efficiencient and do not eliminate the underlying sequential dependency.

In contrast, RQ-MoE achieves parallel decoding through an

*Table 1.* Complexity of encoding and decoding per vector (in FLOPS). For UNQ, $b$ and $H'$ denote codeword dimensionality and hidden layer dimensionality, respectively. For RQ-MoE, we assume $D_e = D$ for simplicity.

|        | Encoding | Decoding |
|--------|----------|----------|
| UNQ    | $H'(D + H + Mb + MK)$ | $H'(b + H' + D + M)$ |
| QINCo  | $2MKD(D + LH)$ | $2MD(D + LH)$ |
| RQ-MoE | $2MKD(D + NLH + N)$ | $2MD(D + NLH + N)$ |

explicit decoupling of the instruction stream and the quantization stream. Crucially, the construction of instruction vectors does not depend on the step-wise dynamic codebooks or intermediate reconstructions. Instead, each instruction vector is updated via Eq. 4, which relies only on the discrete indices and pre-stored expert components. As shown in Fig. 1, this design allows a fast instruction pre-pass, where all instruction vectors $\{\mathbf{I}_1, \ldots, \mathbf{I}_M\}$ are computed via simple lookups and additions and they are independent of the dynamic codebook generation. Once instantiated, the instruction vectors enable the parallel construction of the step-wise specialized codebooks $\{\tilde{\mathcal{C}}^m\}_{m=1}^M$. The final reconstruction $\hat{\mathbf{x}} = \sum_{m=1}^M \tilde{\mathbf{c}}_{i^m}^m$ follows the standard RQ summation, removing the sequential bottleneck present in QINCo. Moreover, this parallelism is not limited to the step level. The second-level MoE transformation within each step can also be executed in parallel across experts, further increasing throughput.

**Complexity.** We further analyze the complexity of encoding and decoding per vector for RQ-MoE, as shown in Tab. 1. Compared to QINCo, the additional computational overhead of RQ-MoE primarily stems from the gating and weighting mechanisms of MoE, which is marginal compared to the total complexity. By enabling step-level parallelism, a theoretical $M\times$ speedup in decoding can be achieved. Furthermore, by incorporating second-level parallelism (i.e., using more experts under a fixed budget with constant $N \times L$), *RQ-MoE can theoretically attain an $N\times$ acceleration in encoding and an $(M \times N)\times$ speedup in decoding.*

### 4.5. Theoretical Analysis

RQ-based quantization methods can be viewed as a sequential decision process that satisfies a Markov property. For standard RQ, the index selection is a memoryless policy conditioned on the current residual at step $m$ ($m \neq 1$):

$$P_{\text{RQ}}(i^m | i^{m-1}, \ldots, i^1) = P_{\text{RQ}}(i^m | x - \sum_{p=1}^{m-1} C_{i^p}^p) \quad (17)$$

$$= P_{\text{RQ}}(i^m | r^m). \quad (18)$$

Dynamic-codebook VQ can be interpreted as reducing the *conditional entropy* in this process:

$$H(i^m | \mathbf{I}^{m-1}, \mathbf{r}^m) \leq H(i^m | \mathbf{r}^m). \quad (19)$$

*Table 2.* Comparison of reconstruction error (MSE) and retrieval recall (R@$k$, %) on four benchmark datasets using a training set of 10M samples. The best results in each category are highlighted in **bold**. We report the quantization fidelity (MSE) and retrieval accuracy (Recall@K) under codebook budgets of 8 and 16 bytes. Note that the MSE values for BigANN and FB-ssnpp are reported in units of $10^4$. For RQ-MoE, we employ $N = 1$ and $L = 16$ by default, and $L = 12$ for the Contriever dataset to align with the QINCo configuration.

| Method | Deep1M ($D = 96$) | | | | BigANN1M ($D = 128$) | | | | FB-ssnpp1M ($D = 256$) | | | | Contriever1M ($D = 768$) | | | |
|---|---|---|---|---|---|---|---|---|---|---|---|---|---|---|---|---|
| | MSE | R@1 | R@10 | R@100 | MSE | R@1 | R@10 | R@100 | MSE | R@1 | R@10 | R@100 | MSE | R@1 | R@10 | R@100 |
| **8 bytes** | | | | | | | | | | | | | | | | |
| OPQ | 0.25 | 15.2 | 51.2 | 87.8 | 2.97 | 21.4 | 63.9 | 95.3 | 9.51 | 2.5 | 5.0 | 11.3 | 1.87 | 8.5 | 24.4 | 50.6 |
| RQ | 0.19 | 22.3 | 64.7 | 95.5 | 2.49 | 27.8 | 75.2 | 98.1 | 9.18 | 2.7 | 5.9 | 14.4 | 1.81 | 9.8 | 27.4 | 53.1 |
| LSQ | 0.17 | 24.6 | 68.7 | 96.8 | 1.89 | 30.9 | 78.5 | 98.8 | 8.82 | 3.4 | 8.0 | 18.0 | 1.64 | 13.3 | 35.1 | 62.9 |
| UNQ | 0.14 | 29.2 | 77.5 | 98.8 | 1.12 | 39.7 | 88.3 | 99.6 | — | — | — | — | — | — | — | — |
| QINCo | 0.12 | 36.3 | 84.6 | 99.4 | 1.12 | 45.1 | 91.0 | 99.7 | 8.67 | 3.6 | 8.9 | 20.7 | 1.42 | 20.4 | 47.1 | 74.5 |
| RQ-MoE | **0.12** | **36.5** | **85.0** | **99.5** | **1.10** | **46.2** | **91.7** | **99.7** | **8.64** | **3.6** | **9.0** | **20.9** | **1.38** | **21.6** | **48.9** | **76.2** |
| **16 bytes** | | | | | | | | | | | | | | | | |
| OPQ | 0.14 | 34.9 | 82.2 | 98.9 | 1.79 | 41.5 | 89.6 | 99.8 | 7.25 | 5.1 | 12.3 | 27.3 | 1.71 | 18.0 | 40.8 | 98.9 |
| RQ | 0.10 | 43.0 | 90.6 | 99.9 | 1.30 | 49.2 | 95.1 | 100.0 | 6.99 | 5.4 | 13.0 | 29.7 | 1.65 | 19.9 | 43.6 | 68.4 |
| LSQ | 0.09 | 42.0 | 89.5 | 99.9 | 0.98 | 51.0 | 95.4 | 100.0 | 6.56 | 6.3 | 16.0 | 34.7 | 1.32 | 25.7 | 54.6 | 79.8 |
| UNQ | 0.06 | 51.5 | 95.8 | 100.0 | 0.47 | 64.3 | 98.8 | 100.0 | — | — | — | — | — | — | — | — |
| QINCo | 0.05 | 59.6 | 98.1 | 100.0 | 0.33 | 71.6 | 99.5 | 100.0 | 6.58 | 6.4 | 16.8 | 35.6 | 1.10 | 31.0 | 61.7 | 85.4 |
| RQ-MoE | **0.05** | **60.2** | **98.6** | **100.0** | **0.30** | **72.3** | **99.6** | **100.0** | **6.53** | **6.5** | **17.0** | **36.1** | **1.08** | **31.4** | **62.5** | **87.2** |

Under this framework, we can prove[1]:

**Theorem 4.1.** *Standard RQ and QINCo can be recovered as constrained special cases of RQ-MoE.*

**Theorem 4.2.** *For any target space $\mathbb{R}^D$ in RQ-MoE, setting the expert dimensionality to $D_e = D$ is sufficient to preserve the information required for contextual manifold-aware codebook adaptation.*

Theorem 4.1 indicates that RQ-MoE is a generalized framework for the dynamic-codebook VQ method, and Theorem 4.2 provides a guideline to set the dimensionality of experts in RQ-MoE, i.e., $D_e \leq D$.

## 5. Experiments

### 5.1. Experimental Setup

**Datasets.** Following prior works (Huijben et al., 2024), we conduct experiments on four large-scale benchmarks containing data from different modalities: Deep1B ($D = 96$) (Babenko & Lempitsky, 2016), BigANN ($D = 128$) (Jégou et al., 2011b), Facebook SimSearchNet++ (FB-ssnpp; $D = 256$) (Simhadri et al., 2021), and Contriever ($D = 768$) (Huijben et al., 2024). Note that we conduct data selection on each benchmark. Hence, their names may be slightly changed according to the data volume, e.g., Deep1M means 1M data in Deep1B. Details are in Appendix B.

**Baselines.** We compare RQ-MoE against diverse baselines. For traditional quantization methods, we use OPQ (Ge et al., 2013), RQ (Chen et al., 2010), and LSQ (Martinez et al., 2018) as baselines. To evaluate neural-based quantization

approaches, we consider UNQ (Morozov & Babenko, 2019) and QINCo (Huijben et al., 2024) as representative baselines. Following prior work (Huijben et al., 2024; Vallaeys et al., 2026), we report the published UNQ results due to the substantial computational overhead associated with its negative mining strategy. QINCo serves as the primary state-of-the-art dynamic-codebook baseline in our comparisons.

**Metrics.** Performance is evaluated using Mean Squared Error (MSE) for reconstruction fidelity and Recall@$k$ ($k \in \{1, 10, 100\}$) for retrieval accuracy, where Recall@$k$ measures whether the true nearest neighbor is retrieved within the top-$k$ results.

**Implementation Details** For traditional quantization methods OPQ, RQ and LSQ, we utilize the high-performance implementations provided by the FAISS library (Douze et al., 2024). For neural-based methods, we follow the experimental settings described in their respective original papers. For RQ-MoE, we set the number of second-level experts to N=1 in the main comparison to isolate the effect of the dual-stream decoupling mechanism without introducing additional model complexity. Across all methods, a single codebook with a size of $K = 256$ is employed. We evaluate performance under two quantization budgets 8 bytes and 16 bytes, and conduct training on two distinct scales: 500K and 10M vectors. More detailed are provided in Appendix B.

### 5.2. Overall Results

As presented in Tab. 2, RQ-MoE achieves leading performance across all metrics and datasets, confirming the effectiveness of our design. Crucially, RQ-MoE consistently matches or outperforms the strongest baseline QINCo,

---

[1]Proofs are provided in Appendix A.1 and Appendix A.2.

*Table 3.* Ablation study on different training scales (500K and 10M). The MSE values for BigANN and FB-ssnpp are reported in units of $10^4$. We evaluate the impact of the NRL (w/o NRL) and the Expert Codebook (w/o $E$).

| | Method | Deep1M | | BigANN1M | | FB-ssnpp1M | |
|---|---|---|---|---|---|---|---|
| | | MSE | R@1 | MSE | R@1 | MSE | R@1 |
| **500K** | RQ-MoE | 0.14 | 31.6 | 1.29 | 42.3 | 8.85 | 3.4 |
| | w/o NRL | 0.15 | 29.2 | 1.54 | 34.7 | 9.03 | 2.9 |
| | w/o $E$ | 0.15 | 29.6 | 1.39 | 39.0 | 9.01 | 2.9 |
| **10M** | RQ-MoE | 0.12 | 36.5 | 1.10 | 46.2 | 8.64 | 3.6 |
| | w/o NRL | 0.14 | 30.7 | 1.26 | 43.4 | 8.86 | 3.4 |
| | w/o $E$ | 0.14 | 31.5 | 1.17 | 44.2 | 8.78 | 3.5 |

*Table 4.* Average encoding and decoding latency per vector (in $\mu$s) on NVIDIA A800 GPUs. For UNQ, $b$ and $H'$ denote codeword dimensionality and hidden layer dimensionality, respectively. For all methods, $M = 8, K = 256, D = 128, H = 256$.

| Method | Setting | Encoding | Decoding |
|---|---|---|---|
| UNQ | $H' = 1024, b = 256$ | 2.3 | 1.5 |
| QINCo | $L = 4$ | 91.8 | 3.3 |
| RQ-MoE | $N = 1, L = 4$ | 96.2 | 1.1 |
| | $N = 2, L = 2$ | 75.4 | 0.7 |
| | $N = 4, L = 1$ | 67.9 | 0.5 |

which is dynamic-codebook but coupled-state method. This empirically validates that a decoupled instruction stream effectively encapsulates the necessary manifold information for conditioning, retaining essential geometric details without the constraint of full reconstruction path dependency. Beyond accuracy, *the robust performance across diverse modalities underscores the versatility of RQ-MoE*. For detailed results on the 500K training scale and further supplementary experiments, please refer to Appendix C.1.

### 5.3. Ablation Study

We evaluate each core component by comparing RQ-MoE against two variants: (1) w/o NRL, which replaces NRL with per-step MSE; and (2) w/o $E$, which removes the expert dimensionality and instead reuses the base codebook $C$ to form the conditioning signal (i.e., duplicate **c** to replace **e** in Eq. 3). Results in Tab. 3 demonstrate that removing NRL leads to a consistent performance degradation, confirming that standard MSE provides insufficient supervision for deep expert networks to capture fine-grained residual geometry. Similarly, the w/o $E$ variant suffers from an information bottleneck; without the independent expert signal, the quantizer fails to adapt dynamically to complex manifold variations, showing the effectiveness of the expert signal. Moreover, the result when $N = 1$ (i.e., no expert) in the experiments reported in Sec. 5.6 also proves that experts indeed improve the reconstruction accuracy.

### 5.4. Analysis of Efficiency

Tab. 4 demonstrates the efficiency of neural methods. We can see that, *while complexity is comparable to QINCo, RQ-MoE significantly accelerates decoding*. At $N = 4, L = 1$, RQ-MoE achieves a 0.5 $\mu$s latency, which is $6.6\times$ faster than QINCo. This speedup is achieved through parallelism at inter-step parallelism via a fast path, and intra-step parallelism within $f_t$ (Eq. 6). Notably, increasing $N$ also leads to a substantial efficiency gain on the encoding side. As $M$ increases to 16, the speedup ratio further expands to over $14\times$; more details are provided in Appendix C.2.

**Comparison to QINCo2.** We notice that the recent work QINCo2 (Vallaeys et al., 2026) achieves acceleration primarily through techniques such as codeword pre-selection. The design of RQ-MoE is orthogonal to QINCo2 and can be naturally combined to further enhance performance.[2]

### 5.5. Analysis of Stream Perturbation

To verify that the quantization and instruction streams play distinct functional roles, we conduct causal perturbation experiments during inference, perturbing one stream at a time while keeping all parameters fixed.

We consider three perturbation strategies: (i) Quantization Perturbation: Replace each selected codeword with the second nearest entry. (ii) Frozen Instruction Accumulation: Freeze the instruction vector after the first three steps. (iii) Non-Accumulated Instruction: Use only the current expert embedding without historical accumulation.

As shown in Fig. 4, perturbing either stream consistently increases reconstruction distortion. The degradation patterns differ: perturbations on the quantization stream cause immediate errors, while perturbations on the instruction stream accumulate over steps, producing more severe errors, particularly when historical accumulation is removed. These results confirm that the two streams serve distinct functional roles and are not interchangeable.

### 5.6. Analysis of Parameter Sensitivity

In Fig. 5, we assess the impact of $N$, $L$, and $D_e$ on reconstruction performance. When the product $N \times L$ is fixed, an excessively large $N$ limits the depth of each expert, while a very small $N$ lacks representation breadth; an optimal $N$ balances depth and breadth to achieve the best accuracy.[3] With $N$ fixed, increasing the network depth $L$ consistently yields performance gains. Furthermore, quantization precision improves as the expert dimensionality $D_e$ increases, but plateaus once $D_e$ exceeds the input dimension $D$. This observation aligns with the theoretical analysis in Sec. 4.5.

---

[2]Experimental comparisons are provided in Appendix C.3.
[3]The complete Pareto frontier is provided in Appendix C.4.

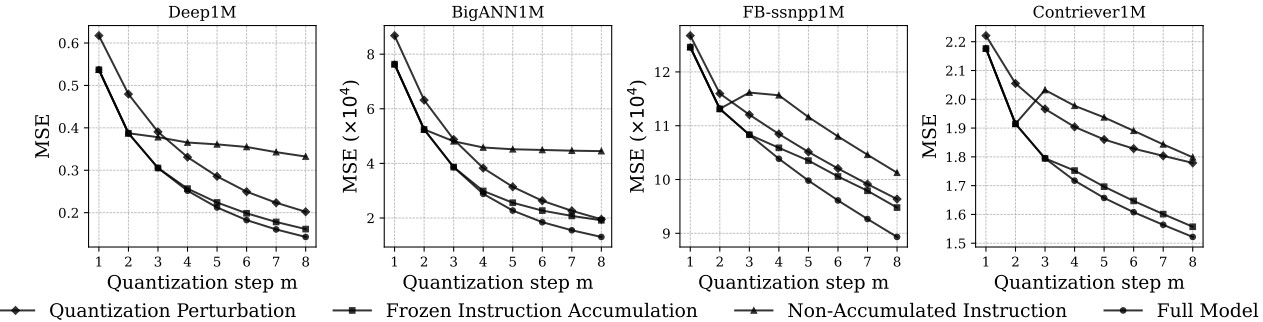

*Figure 4.* Step-wise reconstruction errors under different perturbation strategies on four datasets.

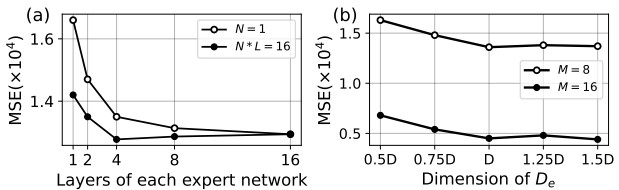

*Figure 5.* Sensitivity of MSE on BigANN1M. (a) Impact of the number of experts $N$ and network depth $L$. (b) Impact of the expert codebook dimension $D_e$ ($N = 2, L = 4$).

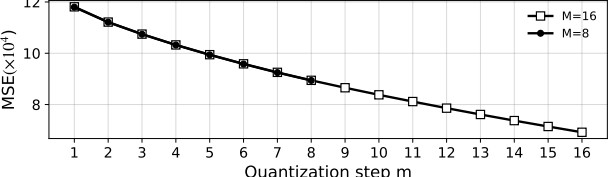

*Figure 6.* MSE for RQ-MoE trained for 8-byte and 16-byte encodings on FB-ssnpp1M, truncated at a varying number of bytes.

### 5.7. Analysis of Dynamic Rates

We evaluate whether a single RQ-MoE model trained for long codes can effectively generate short codes via early termination. Fig. 6 shows MSE per quantization step on BigANN1M for the 16-byte model ($M = 16$), which remains nearly identical to models trained specifically for shorter horizons ($m \leq 8$). This indicates that NRL optimizes deep steps while maintaining the residual hierarchy, preventing subsequent refinements from interfering with earlier parameters. Consequently, this characteristic yields significant practical benefits: it enables compressed domain rate adjustment by simply cropping codes, amortizes training costs by optimizing only for the maximum length $M$, and simplifies model management by maintaining a single master model. Appendix C.5 provides analysis for other datasets.

### 5.8. Analysis of Expert Codebook Gradients

The normalized residual loss computes each step's loss relative to the previous residual, ensuring that contributions from all quantization steps remain on a comparable scale. To investigate its effect on deep quantization layers, we analyze the relative gradient magnitudes of expert codebooks at each quantization step, averaged over training iterations.

Tab. 5 presents the results. Gradients under NRL are consistently higher than those under standard MSE across deeper layers, indicating that NRL provides stronger training signals for later-stage codebooks. In contrast, MSE exhibits rapid gradient decay, which can leave deeper quantization

*Table 5.* Relative gradient magnitudes of expert codebooks at each quantization step. Values are averaged over training iterations and normalized by the gradient of the first layer.

| Loss Type | Quantization Step | | | | | |
|---|---|---|---|---|---|---|
| | 2 | 3 | 4 | 5 | 6 | 7 |
| MSE | 0.812 | 0.568 | 0.212 | 0.073 | 0.017 | 0.008 |
| NRL | 0.837 | 0.613 | 0.382 | 0.247 | 0.108 | 0.042 |

layers under-optimized. Overall, the more gradual decline of gradients under NRL supports more effective training of deep expert codebooks and contributes to improved reconstruction performance.

## 6. Conclusion

In this paper, we present RQ-MoE, a framework combining a two-level MoE with dual-stream quantization to enable input-dependent codebook adaptation for efficient vector quantization. By leveraging a two-level MoE mechanism enabled by implicit routing and NRL, RQ-MoE achieves a substantial acceleration in decoding throughput ($6\times$–$14\times$ faster) while maintaining superior reconstruction fidelity across diverse datasets. Theoretically, we establish RQ-MoE as a generalized framework that unifies existing RQ methods as degenerate cases, and derive a guideline for setting expert dimensionality in RQ-MoE. In the future, we plan to explore how to better adapt RQ-MoE to different applications, e.g., recommender systems and image generation.

## Acknowledgements

This work is supported by the National Key Research and Development Program of China (No. 2025YFE0113500), the National Science Fund for Distinguished Young Scholars (No. 62525605), the National Natural Science Foundation of China (No. 62572410, No. 62272401, and No. U22B2051) and the Natural Science Foundation of Xiamen, China (No. 3502Z202471028).

## Impact Statement

This paper presents work whose goal is to advance the field of machine learning. There are many potential societal consequences of our work, none of which we feel must be specifically highlighted here.

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

# A. Proof for Theorems

## A.1. Proof for Theorem 4.1

**Theorem 4.1.** *Standard RQ and QINCo can be recovered as constrained special cases of RQ-MoE.*

*Proof.* RQ-based quantization methods satisfy a Markov property. For RQ-MoE at step $m$, let the state be $\mathbf{s}^m = (\mathbf{r}^m, \mathbf{I}^m)$, where $\mathbf{r}^m$ is the residual and $\mathbf{I}^m$ is the accumulated instruction. The action $a^m$ corresponds to selecting an index $i^m$. The policy of RQ-MoE is defined as the conditional probability of selecting an index given the current joint state:

$$P_{\text{RQ-MoE}}(i^m | i^{m-1}, \dots, i^1) = P_{\text{RQ-MoE}}(i^m | \mathbf{r}^m, \mathbf{E}_{i^{m-1}}^{m-1}, \dots, \mathbf{E}_{i^1}^1) \tag{20}$$

$$= P_{\text{RQ-MoE}}(i^m | \mathbf{r}^m, f_{ins}(\mathbf{I}^{m-1}, \mathbf{E}_{i^{m-1}}^{m-1})) \tag{21}$$

$$= P_{\text{RQ-MoE}}(i^m | \mathbf{r}^m, \mathbf{I}^m). \tag{22}$$

The state transition for the instruction stream is $\mathbf{I}^m = f_{ins}(\mathbf{I}^{m-1}, \mathbf{E}_{i^{m-1}}^{m-1})$, and the residual transition is $\mathbf{r}^{m+1} = \mathbf{r}^m - f_t(\mathbf{c}_{i^m}^m, \mathbf{I}^m)$. We now demonstrate that imposing specific constraints on the instruction space $\mathcal{I}$ and the expert mapping recovers the policies of Standard RQ and QINCo.

### Case 1: Recovery of Standard RQ (Memoryless Policy)
Standard RQ utilizes static codebooks independent of the quantization history. This corresponds to the case where the instruction stream is nullified. Let the expert dimension $D_e = 0$ (or equivalently, $\mathbf{e}_k^m = \mathbf{0}$ for all $k, m$). Under this constraint, the instruction vector becomes a constant zero vector $\mathbf{I}^m = \mathbf{0}, \forall m$. Consequently, the transformation function becomes an identity mapping $f_t(\mathbf{c}, \mathbf{0}) = \mathbf{c}$. The policy in Eq. 20 degenerates to:

$$P_{\text{RQ-MoE}}(i^m | i^{m-1}, \dots, i^1) = P_{\text{RQ-MoE}}(i^m | \mathbf{r}^m, \mathbf{0}) \tag{23}$$

$$\equiv P_{\text{RQ}}(i^m | \mathbf{r}^m). \tag{24}$$

For RQ, since the $f_t$ is a identity mapping, the residual transition becomes $\mathbf{r}^{m+1} = \mathbf{r}^m - \mathbf{c}_{i^m}^m$. This recovers the memoryless policy of Standard RQ, where the action depends solely on the current residual.

### Case 2: Recovery of QINCo (Coupled-State Policy)
QINCo conditions codebook generation on the cumulative reconstruction value $\hat{\mathbf{x}}^m$, representing a strict coupling in our framework. By setting the expert component identical to the base codeword ($\mathbf{e}_k^m \equiv \mathbf{c}_k^m$ for all $k, m$) and defining the transformation function as the deep residual MLP $f_\theta$ used in QINCo ($f_t \equiv f_\theta$), the update rule simplifies to:

$$\mathbf{I}^m = f_{ins}(\mathbf{I}^{m-1}, \mathbf{E}_{i^{m-1}}^{m-1}) \tag{25}$$

$$= \mathbf{I}^{m-1} + f_t(\mathbf{c}_{i^m}^m, \mathbf{I}^{m-1}) \tag{26}$$

$$= \mathbf{I}^{m-1} + \tilde{\mathbf{c}}_{i^m}^m \tag{27}$$

$$= \hat{\mathbf{x}}^m. \tag{28}$$

Substituting this into Eq. 21, the policy becomes:

$$P_{\text{RQ-MoE}}(i^m | i^{m-1}, \dots, i^1) = P_{\text{RQ-MoE}}(i^m | \mathbf{r}^m, f_{ins}(\mathbf{I}^{m-1}, \mathbf{E}_{i^{m-1}}^{m-1})) \tag{29}$$

$$\equiv P_{\text{QINCo}}(i^m | \mathbf{r}^m, \hat{\mathbf{x}}^m) \tag{30}$$

$$= P_{\text{QINCo}}(i^m | \mathbf{r}^m). \tag{31}$$

Here, the latent control signal $\mathbf{I}^m$ collapses into the reconstruction value $\hat{\mathbf{x}}^m$. Thus, QINCo is identified as a degenerate case where the instruction stream is numerically coupled to the quantization stream.

**Conclusion.** Since both the memoryless policy of Standard RQ and the coupled-state policy of QINCo are derived by placing constraints on $f_t$, $f_{ins}$ and expert codebook $E$, RQ-MoE constitutes a generalized framework for RQ-based quantization. □

## A.2. Proof for Theorem 4.2

**Theorem 4.2.** *For any target space $\mathbb{R}^D$ in RQ-MoE, setting the expert dimensionality to $D_e = D$ is sufficient to preserve the information required for contextual manifold-aware codebook adaptation.*

*Proof.* Let the residual $\mathbf{r}^m \in \mathbb{R}^D$ be supported on a smooth manifold $\mathcal{M}$ with intrinsic dimension $d \leq D$. Since quantization is inherently a finite-precision operation, we analyze the system using the concept of *Metric Entropy*. For a given precision $\epsilon > 0$, let $N_\epsilon(\mathcal{M})$ denote the minimum number of balls of radius $\epsilon$ required to cover $\mathcal{M}$. The entropy at resolution $\epsilon$ is given by $H_\epsilon(\mathcal{M}) = \log N_\epsilon(\mathcal{M})$. For small $\epsilon$, this scales as:

$$H_\epsilon(\mathcal{M}) \approx d \cdot \log(1/\epsilon) + C, \tag{32}$$

where $d$ is the intrinsic dimension.

**Ideal Capacity Formulation.** We model the instruction vector $\mathbf{I}^m \in \mathbb{R}^{D_e}$ as a latent code that must transmit the location of $\mathbf{r}^m$ on the manifold to the codebook generator. Using rate-distortion theory as a conceptual reference, we note that achieving reconstruction error on the order of $\epsilon$ requires the instruction representation to carry information that scales with the metric entropy of the underlying manifold:

$$I(\mathbf{r}^m; \mathbf{I}^m) \geq R(\epsilon) \approx H_\epsilon(\mathcal{M}). \tag{33}$$

**Bottleneck for $D_e < D$.** Consider the capacity of the instruction space $\mathbb{R}^{D_e}$. The maximum entropy of the instruction vector constrained to a bounded volume at resolution $\epsilon$ scales as:

$$H_\epsilon(\mathbf{I}^m) \approx D_e \cdot \log(1/\epsilon). \tag{34}$$

If $D_e < d \leq D$, comparing the scaling laws (slope of information growth) reveals an information deficit:

$$\lim_{\epsilon \to 0} \frac{H_\epsilon(\mathbf{I}^m)}{H_\epsilon(\mathcal{M})} = \frac{D_e}{d} < 1. \tag{35}$$

This implies that as the quantization precision requirements increase ($\epsilon \to 0$), the instruction stream $\mathbf{I}^m$ lacks the sufficient degrees of freedom to distinguish distinct points on $\mathcal{M}$. This *dimensional bottleneck* prevents the instruction stream from faithfully encoding local manifold variations and empirically manifests as degraded reconstruction fidelity.

**Sufficiency for $D_e = D$.** Although the exact intrinsic dimension $d$ is latent and input-dependent (typically estimable via empirical ablation), $D$ serves as a guaranteed upper bound. When $D_e = D \geq d$, the instruction space possesses sufficient dimensional capacity to admit a locally expressive parametrization that preserves the essential degrees of freedom required for manifold-aware adaptation of the manifold $\mathcal{M}$. Consequently, the capacity of the instruction channel scales sufficiently with the source entropy:

$$H_\epsilon(\mathbf{I}^m) \sim H_\epsilon(\mathcal{M}). \tag{36}$$

Thus, setting $D_e = D$ ensures that the instruction vector is sufficient for controlling manifold-aware codebook adaptation in terms of information dimension, allowing the system to reduce distortion without being limited by the capacity of the instruction representation.

**Conclusion.** Therefore, $D_e = D$ constitutes a sufficient dimensionality for the instruction stream under finite-precision quantization. Increasing $D_e$ beyond $D$ adds redundant capacity that exceeds the intrinsic information growth rate of the source residual $\mathbf{r}^m$. $\qquad\square$

## B. Detailed Experimental Settings

### B.1. Descriptions of Datasets

To evaluate the performance of RQ-MoE across diverse data distributions and vector dimensions, we conduct experiments on four large-scale benchmarks:

1. Deep1B ($D = 96$) (Babenko & Lempitsky, 2016) contains 1 billion image descriptors extracted using a pre-trained GoogLeNet model and is a standard benchmark for vector quantization.

2. BigANN ($D = 128$) (Jégou et al., 2011b) is a classic benchmark consisting of 1 billion SIFT descriptors, representing handcrafted local features.

3. Facebook SimSearchNet++ (FB-ssnpp; $D = 256$) (Simhadri et al., 2021) comprises over 1 billion image embeddings optimized for image copy detection, reflecting the data distribution in modern web search infrastructures.

4. Contriever ($D = 768$) (Huijben et al., 2024) consists of 21 million 100-token passage embeddings extracted from Wikipedia. It represents the high-dimensional latent space typical of modern NLP and dense passage retrieval tasks.

## B.2. Data Partitioning and Specifications

For a consistent evaluation across all datasets, we adhere to a rigorous partitioning and configuration protocol:

- **Data Selection**: For each dataset, we extract the first 500,000 (500K) and 10,000,000 (10M) vectors from the original training split for our training sets.

- **Validation Set**: We reserve the 10,000 vectors immediately following the training segment as a validation set to monitor performance and implement the early stopping criterion.

- **Test Set and Evaluation**: Both reconstruction error (e.g., Mean Squared Error) and retrieval performance (e.g., Recall@N) are evaluated on the first 1,000,000 (1M) vectors of the test set.

- **Codebook Configuration**: Across all methods and stages, the number of entries in each codebook is strictly controlled at $K = 256$.

- **Dimensionality**: All quantization processes are conducted in the original feature space of the datasets without any dimensionality reduction such as PCA. For instance, we maintain the original 128-D for BigANN1M and 96-D for deep1M.

## B.3. Optimization and Convergence

All neural models are implemented in PyTorch and optimized using the Adam optimizer. We maintain a constant learning rate of $1 \times 10^{-3}$ and a batch size of $1,024$. To ensure optimal generalization and avoid unnecessary computation, we utilize an early stopping mechanism: training is automatically terminated if the validation loss fails to decrease for 10 consecutive epochs. The model parameters associated with the lowest recorded validation loss are preserved for final evaluation. Latency is measured using a batch size of 4,096 with 10 warm-up iterations and 100 synchronized runs.

## B.4. Computational Infrastructure

The experiments are conducted on a high-performance server equipped with $4 \times$ RTX 3090 GPUs. The host system is powered by a single Intel Xeon Gold 6330 CPU (2.00 GHz, 28 physical cores), which facilitates high-throughput data orchestration and parallel preprocessing.

## B.5. Training Strategy and Initialization

To optimize training efficiency and accelerate model convergence, we perform preliminary training using RQ (Chen et al., 2010) implementation provided by the FAISS library (Douze et al., 2024). The codebooks obtained from this stage serve as a robust initialization for our subsequent process.

# C. Supplementary Experiments

In this section, we present additional experimental results that were omitted from the main text due to the page limit.

## C.1. Performance on Different Data Scales

In this experiment, we provide a comprehensive summary of the reconstruction accuracy and retrieval performance under two data scales: 500K and 10M vectors. The evaluations are conducted across two quantization budgets, specifically 8-byte (8 layers) and 16-byte (16 layers) configurations. Throughout these trials, the hyperparameters are held constant at $H = 256$, $N = 1$, and $L = 16$. Tab. 6 summarizes the comparative results, demonstrating the robustness and scalability of our proposed method in terms of both quantization precision and search efficiency. The results presented here lead to several key observations.

*Table 6.* Comparison of reconstruction error (MSE) and retrieval recall (R@$k$, %) on four benchmark datasets using a training set of 500K and 10M samples. The best results in each category are highlighted in **bold**. We report the quantization fidelity (MSE) and retrieval accuracy (Recall@$K$) under codebook budgets of 8 and 16 bytes. Note that the MSE values for BigANN and FB-ssnpp are reported in units of $10^4$. For RQ-MoE, we employ $N = 1$ and $L = 16$ by default, and $L = 12$ for the Contriever dataset to align with the QINCo configuration.

| | Deep1M ($D = 96$) | | | | BigANN1M ($D = 128$) | | | | FB-ssnpp1M ($D = 256$) | | | | Contriever1M ($D = 768$) | | | |
|---|---|---|---|---|---|---|---|---|---|---|---|---|---|---|---|---|
| Method | MSE | R@1 | R@10 | R@100 | MSE | R@1 | R@10 | R@100 | MSE | R@1 | R@10 | R@100 | MSE | R@1 | R@10 | R@100 |
| | | | | | | | | **8 bytes (500K)** | | | | | | | | |
| OPQ | 0.26 | 16.0 | 51.3 | 88.6 | 2.95 | 21.7 | 64.3 | 95.2 | 9.52 | 2.5 | 5.1 | 11.0 | 1.87 | 8.1 | 24.5 | 50.9 |
| RQ | 0.20 | 21.3 | 63.6 | 95.2 | 2.49 | 27.5 | 75.4 | 98.2 | 9.20 | 2.7 | 6.1 | 13.7 | 1.82 | 10.4 | 27.0 | 52.8 |
| LSQ | 0.17 | 24.5 | 69.4 | 96.9 | 1.91 | 31.7 | 79.4 | 98.9 | 8.87 | 3.3 | 7.5 | 17.3 | 1.65 | 13.1 | 33.9 | 62.7 |
| UNQ | 0.16 | 26.7 | 72.6 | 97.3 | 1.51 | 34.6 | 82.8 | 99.0 | — | — | — | — | — | — | — | — |
| QINCo | 0.15 | 29.4 | 77.6 | 98.5 | 1.40 | 39.5 | 86.4 | 99.3 | 9.01 | 2.9 | 7.5 | 16.8 | 1.60 | 15.1 | 36.7 | 63.5 |
| RQ-MoE | **0.14** | **31.6** | **79.2** | **98.9** | **1.29** | **42.3** | **91.1** | **99.7** | 8.85 | 3.4 | 7.9 | 17.6 | 1.52 | 16.0 | 41.3 | 67.4 |
| | | | | | | | | **16 bytes (500K)** | | | | | | | | |
| OPQ | 0.14 | 34.9 | 82.2 | 98.9 | 1.79 | 40.5 | 89.9 | 99.8 | 7.25 | 5.0 | 11.8 | 25.9 | 1.71 | 18.3 | 40.9 | 65.4 |
| RQ | 0.10 | 43.0 | 90.8 | 99.8 | 1.30 | 49.0 | 95.0 | 100.0 | 7.01 | 5.4 | 13.0 | 29.0 | 1.65 | 20.2 | 43.5 | 68.2 |
| LSQ | 0.09 | 42.3 | 89.7 | 99.8 | 0.98 | 51.1 | 95.4 | 100.0 | **6.63** | **6.2** | **14.8** | **32.3** | 1.35 | 25.6 | 53.8 | 78.6 |
| UNQ | 0.07 | 47.9 | 93.0 | 99.8 | 0.57 | 59.3 | 98.0 | 100.0 | — | — | — | — | — | — | — | — |
| QINCo | 0.06 | 53.0 | 96.2 | 100.0 | 0.47 | 65.5 | 99.1 | 100.0 | 6.88 | 5.7 | 14.4 | 31.6 | 1.30 | 26.5 | 54.3 | 79.5 |
| RQ-MoE | **0.06** | **53.8** | **96.5** | **100.0** | **0.44** | **66.4** | **99.3** | 100.0 | 6.82 | 6.0 | 14.8 | 32.1 | **1.26** | **27.1** | **55.2** | **80.2** |
| | | | | | | | | **8 bytes (10M)** | | | | | | | | |
| OPQ | 0.25 | 15.2 | 51.2 | 87.8 | 2.97 | 21.4 | 63.9 | 95.3 | 9.51 | 2.5 | 5.0 | 11.3 | 1.87 | 8.5 | 24.4 | 50.6 |
| RQ | 0.19 | 22.3 | 64.7 | 95.5 | 2.49 | 27.8 | 75.2 | 98.1 | 9.18 | 2.7 | 5.9 | 14.4 | 1.81 | 9.8 | 27.4 | 53.1 |
| LSQ | 0.17 | 24.6 | 68.7 | 96.8 | 1.89 | 30.9 | 78.5 | 98.8 | 8.82 | 3.4 | 8.0 | 18.0 | 1.64 | 13.3 | 35.1 | 62.9 |
| UNQ | 0.14 | 29.2 | 77.5 | 98.8 | 1.12 | 39.7 | 88.3 | 99.6 | — | — | — | — | — | — | — | — |
| QINCo | 0.12 | 36.3 | 84.6 | 99.4 | 1.12 | 45.1 | 91.0 | 99.7 | 8.67 | 3.6 | 8.9 | 20.7 | 1.42 | 20.4 | 47.1 | 74.5 |
| RQ-MoE | **0.12** | **36.5** | **85.0** | **99.5** | **1.10** | **46.2** | **91.7** | **99.7** | **8.64** | 3.6 | 9.0 | 20.9 | 1.38 | 21.6 | 48.9 | 76.2 |
| | | | | | | | | **16 bytes (10M)** | | | | | | | | |
| OPQ | 0.14 | 34.9 | 82.2 | 98.9 | 1.79 | 41.5 | 89.6 | 99.8 | 7.25 | 5.1 | 12.3 | 27.3 | 1.71 | 18.0 | 40.8 | 98.9 |
| RQ | 0.10 | 43.0 | 90.6 | 99.9 | 1.30 | 49.2 | 95.1 | 100.0 | 6.99 | 5.4 | 13.0 | 29.7 | 1.65 | 19.9 | 43.6 | 68.4 |
| LSQ | 0.09 | 42.0 | 89.5 | 99.9 | 0.98 | 51.0 | 95.4 | 100.0 | 6.56 | 6.3 | 16.0 | 34.7 | 1.32 | 25.7 | 54.6 | 79.8 |
| UNQ | 0.06 | 51.5 | 95.8 | 100.0 | 0.47 | 64.3 | 98.8 | 100.0 | — | — | — | — | — | — | — | — |
| QINCo | 0.05 | 59.6 | 98.1 | 100.0 | 0.33 | 71.6 | 99.5 | 100.0 | 6.58 | 6.4 | 16.8 | 35.6 | 1.10 | 31.0 | 61.7 | 85.4 |
| RQ-MoE | **0.05** | **60.2** | **98.6** | **100.0** | **0.30** | **72.3** | **99.6** | 100.0 | **6.53** | **6.5** | **17.0** | **36.1** | **1.08** | **31.4** | **62.5** | **87.2** |

**Efficacy of Dynamic Codebooks.** As evidenced by the extensive experimental data, dynamic-codebook Vector Quantization (VQ) methods outperform static-codebook approaches across the vast majority of metrics. This superiority further demonstrates that the distribution of real-world data is seldom isotropic. By analyzing historical quantization information, dynamic codebooks effectively reshape the local manifolds formed by the codebook, leading to significantly more precise quantization boundaries that adapt to data density.

**Comparison with SOTA.** Compared to the current state-of-the-art method, Qinco, RQ-MoE achieves superior or competitive performance across multiple datasets. These results suggest that local manifold information does not necessarily need to be strictly coupled with precise reconstruction vectors. Instead, the supplementary expert codebooks in our RQ-MoE framework can capture local manifold signals through a mechanism decoupled from the primary quantization stream. Furthermore, while maintaining high precision, the dual-stream architecture of RQ-MoE enables accelerated decoding, addressing a common bottleneck in neural-based quantizers.

**Performance on FB-ssnpp and Scalability.** Notably, for the FB-ssnpp dataset trained on 500K vectors with $M = 16$, LSQ exhibits leading performance. We attribute this to the beam search reconstruction strategy employed by LSQ and the inherent simplicity of its architecture, which renders it less susceptible to overfitting on smaller subsets. In contrast, complex neural-based quantization methods may suffer from overfitting when the training scale is limited. However, as the training data is expanded to the 10M scale, the dynamic-codebook methods consistently regain their lead, underscoring the importance of large-scale data for fully realizing the potential of neural-based manifold adaptation.

*Table 7.* Average encoding and decoding latency per vector (in $\mu$s) on NVIDIA A800 GPU. For UNQ, $b$ and $H'$ denote codeword dimensionality and hidden layer dimensionality, respectively. For all methods, $K = 256, D = 128, H = 256$.

| Method | Setting | Encoding | Decoding | Setting | Encoding | Decoding |
|---|---|---|---|---|---|---|
| | | $M = 8$ | | | $M = 16$ | |
| UNQ | $H' = 1024, b = 256$ | 2.3 | 1.5 | $H' = 1024, b = 256$ | 4.4 | 2.9 |
| QINCo | $L = 4$ | 91.8 | 3.3 | $L = 4$ | 342.8 | 10.4 |
| | $N = 1, L = 4$ | 96.2 | 1.1 | $N = 1, L = 8$ | 361.2 | 2.1 |
| RQ-MoE | $N = 2, L = 2$ | 75.4 | 0.7 | $N = 2, L = 4$ | 274.8 | 1.2 |
| | $N = 4, L = 1$ | 67.9 | 0.5 | $N = 4, L = 2$ | 238.3 | 0.7 |

*Table 8.* MSE comparison between QINCo2 and RQ-MoE.

| | Deep1M | | BigANN1M | | FB-ssnpp1M | | Contriever1M | |
|---|---|---|---|---|---|---|---|---|
| Method | w/o BS | w/ BS | w/o BS | w/ BS | w/o BS | w/ BS | w/o BS | w/ BS |
| QINCo2 | 0.15 | 0.12 | 1.40 | 1.13 | 9.01 | 8.56 | 1.60 | 1.44 |
| RQ-MoE | 0.14 | 0.12 | 1.29 | 1.10 | 8.85 | 8.33 | 1.52 | 1.39 |

## C.2. Efficiency with Different Budgets

Tab. 7 reports the average encoding and decoding latency per vector, providing a clear comparison of computational efficiency. Our proposed RQ-MoE demonstrates a significant advancement in decoding efficiency compared to existing neural-based methods.

Specifically, when the number of experts $N$ increases, the encoding stage also benefits from a noticeable speedup, as seen in the transition from $N = 1$ to $N = 4$. The data further reveals that RQ-MoE exhibits more pronounced acceleration when dealing with larger quantization budgets ($M$) and deeper expert network configurations ($L$). As shown in the table:

- For the $M = 8, NL = 4$ configuration, RQ-MoE achieves a decoding speedup of over **6**× compared to the SOTA baseline QINCo ($0.5\mu$s vs. $3.3\mu$s).

- For the $M = 16, NL = 8$ configuration, the performance gap widens further, where our method achieves a remarkable **14.8**× speedup in the decoding phase ($0.7\mu$s vs. $10.4\mu$s).

It is worth noting that the encoding phase also benefits from a consistent speedup as the number of experts $N$ increases. Under a fixed total capacity (constant $NL$), a higher $N$ leads to a shallower depth $L$ for individual expert networks. This reduction in sequential network layers effectively decreases the computational overhead during the manifold-aware feature transformation. As shown in Table 7, in the $M = 16$ setting, the encoding latency is reduced from $361.2\mu$s at $N = 1$ to $238.3\mu$s at $N = 4$.

These results validate that the dual-stream architecture effectively decouples the manifold adaptation from the heavy reconstruction process, enabling high-performance vector quantization that is highly suitable for latency-sensitive retrieval systems.

## C.3. Comparison with QINCo2

We additionally compare RQ-MoE with the recent QINCo2 (Vallaeys et al., 2026) under comparable experimental settings. For consistency, both methods are evaluated with the same compression budget (8 bytes, 500K training subset, and $N \times L = 16$), and beam search is applied where applicable. To focus on the architectural differences between the two methods, we do not enable additional generic acceleration techniques such as candidate pre-selection for either model.

As shown in Tab. 8, RQ-MoE achieves competitive or consistently lower reconstruction error across datasets under both standard and beam-search settings, while exhibiting similar trends in beam-search improvement.

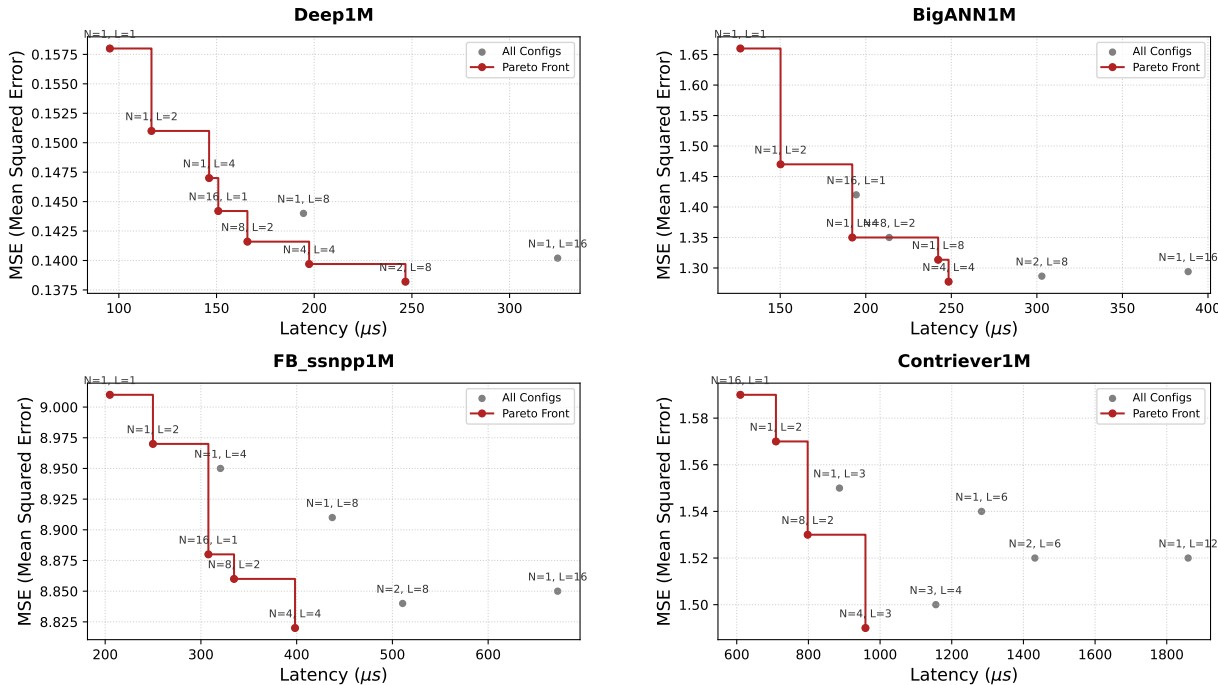

*Figure 7.* Trade-off analysis between MSE and end-to-end inference latency ($\mu s$). Each sub-figure highlights the Pareto-optimal configurations (red line). Points are annotated with their respective hyperparameters (N, L).

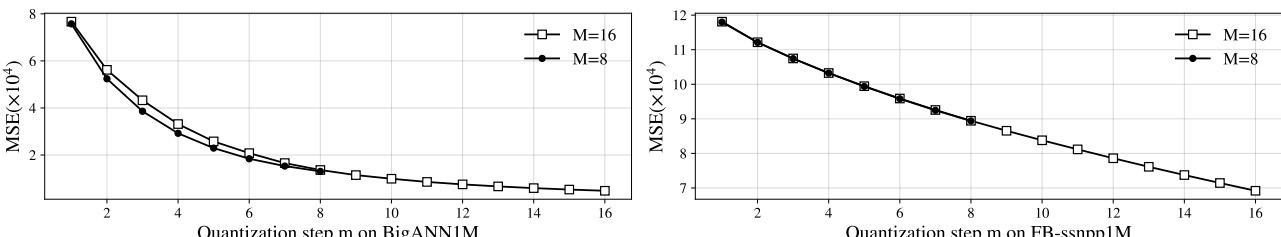

*Figure 8.* MSE for RQ-MoE trained for 8-byte and 16-byte encodings on FB-ssnpp1M and BigANN1M, truncated at a varying number of bytes.

### C.4. Accuracy–Latency Trade-off under Different Expert Configurations

To further analyze the effect of expert specialization, we evaluate RQ-MoE under different numbers of second-level experts while keeping the overall parameter budget comparable. Specifically, we maintain a fixed expert budget by controlling the product of the number of experts and expert depth ($N \times L$), such that the performance gain mainly reflects structured conditional computation rather than increased model capacity.

Fig. 7 presents the accuracy–latency Pareto frontier under different expert configurations on the 500K training setting. The results show that increasing the number of experts consistently improves reconstruction quality by enhancing localized manifold modeling capacity, while introducing additional transformation overhead during decoding. This forms a controllable trade-off between reconstruction accuracy and decoding efficiency.

Importantly, the dual-stream decoupling mechanism already provides strong performance under the $N = 1$ setting, indicating that the primary gain comes from the decoupled instruction–quantization design rather than expert scaling alone. Increasing $N$ further serves as an orthogonal enhancement that improves local adaptive modeling for more complex data distributions.

## C.5. Dynamic Rates on Different Datasets

We further investigate the adaptability of our model across different bit-rates by evaluating its performance under partial decoding scenarios. Fig. 8 illustrates the Mean Squared Error (MSE) per quantization step $m$ for both 8-byte ($M = 8$) and 16-byte ($M = 16$) models.

A key observation from the results is that for both the FB-ssnpp and BigANN datasets, the MSE curves of the 8-byte and 16-byte models are virtually identical for all steps $m \leq 8$. This indicates that a single model trained for a higher quantization budget can be effectively utilized for lower-rate approximations without retraining. Specifically:

- **Data Distribution and Error Convergence**: In Fig. 8 (left), we observe that the MSE for the FB-ssnpp dataset decreases in a nearly linear fashion. This linear trend suggests that the underlying data distribution in FB-ssnpp, which consists of image descriptors for content moderation, is relatively uniform across the manifold. This leads to a consistent reduction in reconstruction error as more quantization levels are added. In contrast, the BigANN dataset (right) shows a more rapid initial decrease, reflecting different local density characteristics inherent in SIFT descriptors.

- **Compressed Domain Rate Adjustment**: The consistency across different $M$ values allows for dynamic rate adjustment. Vectors can be lossily compressed or refined simply by cropping or extending their codes in the compressed domain to meet real-time storage or latency constraints.

- **Amortized Training and Management**: This property implies that the loss at step $m$ has negligible interference with the parameters optimized for steps $< m$. Consequently, a single $M = 16$ model can serve as a universal quantizer for any $m \leq 16$, significantly reducing the computational cost of training multiple models and simplifying model management in production environments.

