# OpenReview forum: "RQ-MoE: Residual Quantization via Mixture of Experts for Efficient Input-Dependent Vector Compression"
_ICML.cc/2026/Conference — ICML 2026 regular_

### Official Review · Reviewer_jUvC · 2026-03-11

**Soundness:** 3
**Presentation:** 3
**Significance:** 3
**Originality:** 3
**Overall Recommendation:** 4
**Confidence:** 4

**Summary:**

The authors examine the problem of vector compression through residual quantization enhanced with a Mixture-of-Experts architecture.
They investigate a general aspect of input-dependent vector quantization by introducing RQ-MoE, a framework that integrates a two-level MoE mechanism with dual-stream quantization.
The method dynamically adapts codebooks based on input geometry while enabling parallel decoding.
Experiments on several large-scale embedding datasets show competitive reconstruction and retrieval performance with significantly faster decoding compared to existing methods.

**Compliance With Llm Reviewing Policy:**

Affirmed.

**Final Justification:**

The rebuttal has addressed most of my initial concerns, and thus I support the publication.

**Key Questions For Authors:**

1. **Effect of Expert Capacity.**
The proposed framework introduces experts in the second-level MoE to adapt the codebooks dynamically.
How sensitive is the performance to the number of experts and the expert network capacity under a fixed parameter or compute budget?
A controlled comparison with equivalent-capacity non-MoE models would clarify whether the gain mainly comes from conditional computation or simply increased model capacity.

2. **Encoding Efficiency in Practical Systems.**
The paper emphasizes decoding acceleration through parallelization.
However, the encoding stage still follows a sequential residual refinement similar to standard RQ.
In practical retrieval systems where encoding latency also matters (e.g., online indexing), how does the encoding cost compare to prior neural quantizers when evaluated end-to-end?

3. **Ablation on the Dual-Stream Design.**
The method relies on a dual-stream architecture that separates the instruction stream from the quantization stream.
An ablation isolating this design choice would help clarify its contribution.
For example, how does performance and efficiency change if the instruction stream is removed or replaced with a simpler conditioning mechanism?
This question aligns with **W2** written above; more clarification on design choices, particularly in theoretical, conceptual, and empirical perspectives should be required.

**Recommendation Justification**

The paper proposes an interesting framework combining residual quantization with Mixture-of-Experts and demonstrates promising efficiency improvements.
However, several aspects remain unclear, particularly the motivation for employing MoE in this context and the limited experimental validation of the design choices.
Additional experiments and explanations would help clarify the generality and necessity of the proposed method.
I am open to raising the score if these concerns are adequately addressed in the rebuttal.

**Limitations:**

No. The limitations and social impact of the work are not adequately covered within the manuscript.
This should be addressed during the rebuttal period.

**Strengths And Weaknesses:**

**Strengths**

1. **Novelty.**
The paper proposes a novel integration of residual quantization with a two-level MoE mechanism and a dual-stream quantization design, enabling input-dependent codebook adaptation without additional bit overhead.

2. **Efficiency Improvements.**
The proposed architecture decouples instruction generation from quantization, enabling parallel decoding.
Empirical results indicate substantial decoding acceleration compared to dynamic-codebook baselines.

3. **Comprehensive Empirical Evaluation.**
Experiments cover multiple large-scale datasets with varying embedding dimensions, where RQ-MoE demonstrates competitive reconstruction and retrieval performance while improving inference efficiency.

**Weaknesses**

1. **Motivation for Applying MoE to Residual Quantization.**
The paper introduces a Mixture-of-Experts design inside residual quantization, yet the necessity of MoE in this context remains somewhat unclear.
While MoE increases representational flexibility, the paper does not convincingly explain why such conditional computation is particularly suitable or necessary for RQ-based vector compression.

2. **Limited Experimental Scope on Design Choices.**
The evaluation primarily compares against existing quantization baselines but does not sufficiently explore alternative architectural choices or simplified variants that could isolate the actual benefit of the MoE component.

3. **Clarity of Theoretical Contribution.**
The theoretical claim that existing methods are degenerate cases of the proposed framework is interesting, but the practical implications of this result remain somewhat unclear and are not empirically validated.

---

> ### Author Rebuttal · Authors · 2026-03-31
>
> Dear Reviewer jUvC:
>
> We sincerely thank the reviewer for the constructive feedback and for recognizing the originality and performance of RQ-MoE. In response to the specific points raised in your review, we provide the following clarifications.
>
> **Q1: Motivation and Necessity of MoE in Residual Quantization.**
>
> The fundamental motivation for integrating MoE into residual quantization (RQ) is to achieve input-dependent codebook adaptation, a goal that aligns perfectly with the core philosophy of MoE systems.
>
> In fact, standard RQ can be viewed as a degenerate instance of an MoE architecture, where Euclidean distance serves as the routing mechanism and the selected codeword acts as a Top-1 expert. We extend this intuition by reusing the gate and index from the quantization process to implicitly represent the first-level MoE, which enables our unique dual-stream construction without additional overhead.
>
> For local manifold fitting, MoE effectively combines various basis vectors (experts) to characterize local subspaces with high precision. Compared to a monolithic long MLP, our MoE-based approach achieves internal architectural parallelization and mitigates the risk of overfitting often associated with deep, non-conditional networks. By decomposing the task into region selection and specialized local modeling, RQ-MoE captures complex input geometries more effectively than static or globally transformed codebooks.
>
>
> **Q2: Model Capacity under Controlled Budget.**
>
> We carefully control both computation and parameter budgets. The first-level MoE introduces no extra bits via index reuse, and the second-level MoE keeps the total budget fixed by maintaining constant N×L, ensuring comparable parameter counts across models.
>
> Thus, increasing the number of experts redistributes capacity into conditional components rather than increasing total capacity. The performance gain therefore comes from structured conditional computation rather than parameter scaling. Additional results under different budgets are provided in Fig.6: https://anonymous.4open.science/r/Rebuttal-to-RQMoE.
>
> **Q3: Ablation on dual-stream design.**
>
> The dual-stream architecture enables parallel decoding by decoupling the quantization stream from the instruction stream. To verify its necessity, we conducted ablations where the instruction stream was replaced with simpler conditioning signals. As shown by the "w/o E" setting in Figure 4, using a basic accumulation of base codewords leads to a significant degradation in accuracy. We further tested a "Base + Offset" variant, which similarly underperformed. These variants lead to degraded reconstruction performance.
>
> |Method|Deep1M|BigANN1M|FB-ssnpp1M|
> |-|-|-|-|
> Base (w/o E)|0.15|1.39|9.01|
> Base + Offset|0.14|1.31|8.93|
> RQ-MoE|0.14|1.29|8.85|
>
> **Q4: Encoding Efficiency.**
>
> We emphasize the asymmetry between encoding and decoding in large-scale retrieval systems: encoding is performed once per vector, while decoding is repeatedly executed at query time, making decoding efficiency dominant in practice. Nevertheless, encoding latency is important in scenarios such as online indexing. In our framework, encoding cost mainly comes from codebook transformation and can be reduced via lightweight preselection based on distances to the base codebook. On BigANN1M, combining top-A (A=16) with beam search (beam size = 16) yields an 8× speedup (96.2$\mu$s to 11.4$\mu$s) with only marginal MSE degradation (1.29 to 1.31).
>
> **Q5: Practical Implications of the Theoretical Framework.**
>
> The primary objective of our theoretical framework is not merely to provide a unified formulation, but to establish a principled foundation for achieving performance superior to existing methods like QINCo. By demonstrating that standard RQ and QINCo are constrained, degenerate cases of our model, the framework reveals a structured design space where standard RQ lacks conditioning and QINCo relies on a fully coupled conditioning-reconstruction design. This coupling in prior methods creates an inherent "information entanglement" that restricts the model's capacity to optimize the instruction and quantization streams independently.
>
> Our theoretical results indicate that decoupling these processes through a dual-stream architecture is the key to unlocking higher representational precision. This perspective directly motivates our design: by separating the instruction stream from the reconstruction process, we not only enable parallel decoding but also provide a more flexible optimization landscape that allows the model to capture more complex manifold features than QINCo. Therefore, the theoretical framework serves as a justification for the necessity of our dual-stream quantization, proving that such a decoupled architecture is essential for both structural efficiency and state-of-the-art accuracy. We will further elaborate on these practical implications and their impact on performance in the revised manuscript.

---

> > ### Author Rebuttal · Reviewer_jUvC · 2026-04-01
> >
> > Thanks for the detailed rebuttal.
> > I appreciate the time and effort you put into addressing my concerns.
> > Overall, the rebuttal has addressed most of my concerns, where I am now inclined to support the publication.

---

> > > ### Author Response · Authors · 2026-04-06
> > >
> > > Dear Reviewer jUvC:
> > >
> > > Thank you very much for your kind and encouraging feedback. We truly appreciate your time and the thoughtful evaluation of our work. We are glad that our rebuttal has addressed your concerns and that you find the work more convincing.
> > >
> > > We are truly grateful for your professional support. Your recognition of our efforts and your support mean a lot to us.

---

### Official Review · Reviewer_E6Er · 2026-03-11

**Soundness:** 2
**Presentation:** 2
**Significance:** 2
**Originality:** 2
**Overall Recommendation:** 5
**Confidence:** 5

**Summary:**

This paper presents a residual quantization method based on a mixture-of-experts framework that generates dynamic codebooks. While the baseline method, QINCo, suffers from sequential decoding, the proposed framework generates dynamic codebooks in parallel by utilizing instruction vectors computed from expert codebooks. In addition, the paper proposes a normalized-residual loss that replaces the conventional mean squared error and stabilizes the training of residual quantization.

**Compliance With Llm Reviewing Policy:**

Affirmed.

**Final Justification:**

The rebuttal and additional analyses have addressed my main concerns sufficiently to change my assessment. In particular, the additional evidence makes the use of the instruction stream for conditioning more convincing, suggesting that the benefit of parallel decoding does not simply come from using a weaker conditioning signal, but from a functionally meaningful decoupled design. The additional analysis of NRL also strengthens the empirical support for the proposed training objective.

**Key Questions For Authors:**

1. Could the authors provide an experimental comparison between MSE and NRL in terms of training stability, such as convergence behavior, gradient magnitude, or stability across quantization steps?
2. Could the authors provide further analysis of whether the expert codebook captures information that is complementary to the base codebook, rather than largely redundant with it?
3. Could the authors evaluate how QINCo would perform if its conditioning signal, i.e., the accumulation of previously generated dynamic codewords, were replaced with the sum of base codewords?
4. Have the authors evaluated how QINCo performs with NRL? NRL appears likely to benefit other RQ-based methods as well.
5. NRL down-weights samples with large residual norms. Could the authors clarify whether its benefit comes from preventing overfitting to outliers, or partly from reducing the contribution of hard (high distortion) samples, which may be insufficiently optimized?

**Limitations:**

yes

**Strengths And Weaknesses:**

[Strengths]
1. The proposed framework enables parallel decoding in residual quantization with dynamic codebooks, thereby reducing decoding latency.
2. The normalized-residual loss significantly improves the performance compared to the MSE loss.
3. The proposed methods achieve the state-of-the-art performances on the benchmarks.

[Weaknesses]
1. The paper provides insufficient analysis of the normalized-residual loss. The paper claims that MSE-based training for residual quantization is unstable, but a more direct experimental comparison between MSE and NRL would be necessary to support this claim.
2. While the paper suggests that the expert codebook encodes local manifold characteristics of the input, it remains unclear whether the expert codebook learns information that is complementary to the base codebook or largely overlaps with it.
3. One of the main concerns is that the parallel decoding advantage of RQ-MoE may come from conditioning on a weaker historical signal. In QINCo, the dynamic codebook at step $m$ depends on the partial reconstruction from previous steps, i.e., the accumulation of previously generated dynamic codewords, which is why decoding remains sequential. If one replaced QINCo's conditioning with a weaker signal, such as a sum of the base codewords, one might also remove the sequential dependency, but at the cost of using less informative conditioning for dynamic codebook generation.
4. Although RQ-MoE reduces decoding latency through parallel decoding, QINCo's sequential decoding already takes only 3.3 μs. Moreover, the ablation study in Table 4 suggests that, when focusing on the architectural effect alone (i.e., w/o NRL), QINCo is actually stronger than RQ-MoE. This raises the question of whether it is worth trading off accuracy for lower decoding latency in this setting.

---

> ### Author Rebuttal · Authors · 2026-03-31
>
> Dear Reviewer E6Er:
>
> We thank the reviewer for the constructive feedback. We address the key concerns below.
>
> **Q1: The effectiveness and analysis of NRL.**
>
> NRL addresses the scale imbalance across quantization steps. Under MSE, later-stage residuals have much smaller magnitudes, leading to vanishing gradients and unstable multi-stage optimization. NRL normalizes residuals relative to previous steps, ensuring balanced gradient contributions. It also exhibits bounded (redescending) gradients, reducing sensitivity to extreme residuals.
>
> We further compare MSE and NRL in terms of convergence and gradient norms across steps. The results show that NRL leads to significantly improves gradient flow, particularly for the instruction codebooks. Compared to MSE, deeper quantization stages receive substantially larger gradient norms under NRL. This indicates more effective training of later-step codebooks, which are typically under-optimized with MSE, and leads to better utilization of deep quantization steps and improved reconstruction accuracy. Detailed experimental results are provided in Fig. 1–5 at https://anonymous.4open.science/r/Rebuttal-to-RQMoE.
>
> **Q2: Whether expert codebooks provide complementary information.**
>
> We empirically verify the role of expert codebooks through ablations. The w/o E setting, reported in Table 4 of the main paper, replaces the expert codebooks with the base codebook and shows a clear performance drop. We also explored an alternative design in earlier experiments where expert codebooks are parameterized as offsets on top of the base codebook, i.e., a “floating basis”. This variant also consistently underperforms compared to independently learned expert codebooks. These results indicate that expert codebooks provide complementary modeling capacity that cannot be trivially derived from the base codebook.
>
> |Method|Deep1M|BigANN1M|FB-ssnpp1M|
> |-|-|-|-|
> base (w/o E)|0.15|1.39|9.01|
> base + offset|0.14|1.31|8.93|
> RQ-MoE|0.14|1.29|8.85|
>
> Using base codewords alone leads to a mismatch between conditioning signals and the quantization trajectory. Instead, our design encourages expert codebooks to act as high-dimensional carriers of residual and manifold-aware signals, progressively accumulating information that guides subsequent quantization steps. This supports a complementary functional role: base codebooks focus on reconstruction, while expert codebooks encode conditioning signals. This can also be interpreted as encoding the quantization path rather than the reconstructed vector itself (as in QINCo). Overall, these results confirm that expert codebooks encode complementary information rather than redundant variations of the base codebook.
>
> **Q3: Whether parallel decoding relies on weaker conditioning.**
>
> In QINCo, conditioning is entangled with reconstruction, requiring codewords to serve both roles, which can lead to suboptimal trade-offs. In contrast, RQ-MoE explicitly decouples these roles. The quantization stream focuses purely on reconstruction, while the instruction stream accumulates historical information for conditioning via expert codebooks. This decoupling allows conditioning signals to evolve independently of reconstruction, preserving rich contextual information without introducing sequential dependency. Therefore, the conditioning signal is not weaker, but more flexible and better aligned with its role.
>
> **Q4: The role of NRL and its interaction with QINCo.**
>
> We have evaluated QINCo trained with NRL and observe limited improvement compared to its standard formulation. We attribute this to the fact that in QINCo, the conditioning signal is strongly constrained, as it must coincide with the reconstructed codewords. This restricts the optimization space, reducing the impact of improved supervision such as NRL.
>
> In contrast, RQ-MoE introduces a decoupled instruction stream that is not tied to reconstruction. This additional flexibility increases both representational capacity and optimization difficulty, making stable and balanced supervision more critical. NRL plays a key role by normalizing residual scales across steps, particularly for deeper quantization stages.
>
> |Method|Deep1M|BigANN1M|FB-ssnpp1M|Contriever1M|
> |-|-|-|-|-|
> QINCo w/o NRL|0.15|1.40|9.01|1.60|
> QINCo w/ NRL|0.14|1.33|9.04|1.56|
> RQ-MoE w/ NRL|0.14|1.29|8.85|1.52|
>
> These results suggest that while NRL is generally applicable, it is particularly beneficial in RQ-MoE, where the model has greater representational freedom but also requires stronger optimization guidance.
>
> Finally, regarding the trade-off between accuracy and decoding latency, our full model (with NRL) achieves SOTA or comparable accuracy while providing substantial decoding speedups, indicating that the trade-off does not arise in the final model. This indicates that the gains of RQ-MoE cannot be attributed to NRL alone, but arise from the combination of the proposed architecture and improved optimization.

---

> > ### Author Rebuttal · Reviewer_E6Er · 2026-04-03
> >
> > I appreciate the authors' responses to my concerns. However, I still have reservations about the use of the instruction stream for conditioning. My further questions are as follows.
> >
> > 1. In QINCo, the conditioning signal is produced by repeated MLP transformations, whereas in RQ-MoE it is formed by the summation of learned parameters. Could the authors provide empirical evidence that this simpler signal is not a weaker surrogate for QINCo's conditioning signal?
> >
> > 2. Could the authors provide more direct empirical evidence that decoupling into quantization and instruction streams is functionally meaningful, e.g., by perturbing one stream and measuring the change in distortion?
> >
> > 3. The comparison between QINCo w/ NRL and RQ-MoE in the rebuttal appears to have been conducted using 500K training vectors. Could the authors provide the same comparison under the 10M training setting as well?
> >
> > Clarifying these points would help me better assess the validity of the authors' claims and the contribution of the proposed method.

---

> > > ### Author Response · Authors · 2026-04-06
> > >
> > > Dear Reviewer E6Er:
> > >
> > > We sincerely thank the reviewer for the constructive follow-up questions and for the time spent evaluating our work. These questions help us further clarify the design choices and provide more direct empirical evidence regarding the proposed dual-stream architecture. We address each point below.
> > >
> > > **Q1: Is summation-based signal weaker than MLP-based signal?**
> > >
> > > To directly address this concern, we replace the instruction update in RQ-MoE with a QINCo-style MLP-based formulation: $I^m = MLP([I^{m-1}; E^{m-1}_{i^{m-1}}])$, where MLP is implemented as a four-layer network with ReLU and residual connections. All other components are kept identical for a controlled comparison. The reconstruction errors (MSE) are reported below.
> > >
> > > |Method|Deep1M|BigANN1M|FB-ssnpp1M|Contriever1M|
> > > |-|-|-|-|-|
> > > |sum|0.14|1.29|8.85|1.52|
> > > |MLP|0.14|1.30|8.84|1.54|
> > >
> > > The results show that MLP-based signal brings negligible improvement in reconstruction accuracy while significantly increasing training time.
> > >
> > > We interpret this as follows. (1) The instruction stream mainly accumulates contextual information along the quantization path, for which simple addition is sufficient. (2) The instruction is further processed by deep transformation networks (12/16-layer MLP) to modulate the base codebook, so it only needs to provide a transformable context signal rather than a highly expressive representation.
> > >
> > > Overall, these results indicate that the summation-based formulation is **not a weaker surrogate**, but an efficient and equally effective conditioning mechanism.
> > >
> > > **Q2: Is the decoupled design functionally meaningful?**
> > >
> > > To directly validate the functional role of the decoupled design, we conduct perturbation experiments at inference time, where model parameters are fixed and only one stream is modified at a time. This ensures that the observed changes reflect the causal effect of each stream rather than training adaptation.
> > >
> > > For the quantization stream, we perturb the selected codewords by (I) replacing them with the second nearest entries (based on Euclidean distance) from the base and expert codebooks at each step. This preserves a plausible alternative while removing the optimal routing decision.
> > >
> > > For the instruction stream, we consider two perturbations: (II) using the original instruction update for the first three steps and then freezing it for all subsequent steps; (III) removing accumulation and using only the current expert embedding at each step.
> > >
> > > |Method|Deep1M|BigANN1M|FB-ssnpp1M|Contriever1M|
> > > |-|-|-|-|-|
> > > |I|0.20|1.95|9.63|1.82|
> > > |II|0.16|1.53|9.47|1.59|
> > > |III|0.33|4.45|10.01|1.85|
> > > |RQ-MoE|0.14|1.29|8.85|1.52|
> > >
> > > All perturbations lead to increased distortion, demonstrating that both streams are essential. In particular, removing accumulation (III) causes a substantial degradation, highlighting that the cumulative instruction is critical for capturing the quantization trajectory. Fig. 7 at https://anonymous.4open.science/r/Rebuttal-to-RQMoE further illustrates the detailed step-wise reconstruction errors under various perturbations.
> > >
> > > These results provide direct causal evidence that the decoupled quantization and instruction streams are functionally meaningful, rather than a purely structural design.
> > >
> > > **Q3: Comparison under 10M training setting**
> > >
> > > We further evaluate QINCo with NRL on large-scale (10M) training datasets to provide a direct comparison under the same setting.
> > >
> > > |Method|Deep10M|BigANN10M|FB-ssnpp10M|Contriever10M|
> > > |-|-|-|-|-|
> > > |QINCo w/o NRL|0.12|1.12|8.67|1.42|
> > > |QINCo w/ NRL|0.12|1.12|8.72|1.38|
> > > |RQ-MoE|0.12|1.10|8.64|1.38|
> > >
> > > The results show that NRL brings only limited improvement to QINCo even at the 10M scale, and QINCo still does not outperform RQ-MoE. We attribute this to the strong coupling between the quantization and instruction streams in QINCo. Since the instruction is constrained to coincide with the reconstructed codewords, its representational flexibility is limited. While larger training data allows better parameter optimization, it does not resolve this structural constraint.
> > >
> > > One possible explanation is that QINCo’s conditioning signal primarily reflects the current reconstruction “position”. While intuitive, such a signal may not be the most suitable form for subsequent transformation by deep networks, as it is not explicitly optimized for downstream modulation. From the perspective discussed in Q1, RQ-MoE instead learns an instruction that serves as a context feature tailored for modulating subsequent codebooks, without being tightly coupled to the actual quantization position.
> > >
> > > These results further support that the advantage of RQ-MoE comes from the dual-stream design rather than the training objective alone.
> > >
> > > We hope that these additional experiments and analyses sufficiently address the reviewer’s concerns and clarify the effectiveness and necessity of the proposed design.

---

### Official Review · Reviewer_Xp8D · 2026-03-13

**Soundness:** 2
**Presentation:** 2
**Significance:** 3
**Originality:** 3
**Overall Recommendation:** 3
**Confidence:** 3

**Summary:**

This paper proposes RQ-MoE to address the sequential decoding bottleneck in dynamic codebook vector quantization. The main idea is to decouple the instruction stream from the quantization stream through a dual-stream design. In the first layer, the method uses a high-dimensional codebook with index reuse for implicit routing. In the second layer, it generates dynamic codebooks by conditioning a base codebook on the instruction signal. The paper also introduces normalized residual loss for training. Another claimed contribution is that the framework subsumes both standard RQ and QINCo as special cases. Empirically, the method is reported to match or slightly outperform QINCo in accuracy, while substantially improving decoding efficiency.

**Compliance With Llm Reviewing Policy:**

Affirmed.

**Key Questions For Authors:**

1. Please clarify how the default setting N=1in the main table relates to the MoE naming. It should also be made clear whether the main results validate the mixture mechanism itself, or primarily the dual-stream dynamic quantization design.
2. Please provide a full accuracy–latency Pareto for N=1,2,4 under matched parameter and training budgets. This would help isolate the gain from the experts themselves.
3. Please clarify whether the QINCo degeneration result in the appendix is intended as a strict algorithmic equivalence, or only as a formal reformulation under unified notation.
4. A direct experimental comparison with QINCo2 should be included.
5. The latency evaluation should be described in more detail, including batch size, warm-up, number of runs, and other relevant settings.
6. The method section should clearly specify the dimensional compatibility among D_e, D, and the instruction stream.

**Limitations:**

yes

**Strengths And Weaknesses:**

Strengths:
1. The combination of conditional dynamic codebooks with dual-stream decoding is reasonably novel. The use of index reuse to avoid the overhead of explicit routing is also a sensible design choice. Overall, the paper targets an important problem, and the technical entry point is well motivated.
2. Figures 1 to 3 help illustrate the motivation, the overall architecture, and the differences from PQ and RQ fairly clearly. As a result, the main technical narrative is relatively easy to follow.
3. The experiments cover four datasets. They report both MSE and Recall@1, 10, and 100. The paper also includes efficiency analysis and some ablation studies. Overall, the empirical evaluation is reasonably comprehensive.
4. If the claimed parallel decoding advantage holds in practice, the contribution could be of real value for large-scale vector retrieval systems. Such systems are often constrained by decoding throughput, so improving this aspect would have clear practical significance.

Weaknesses:
1. There is a mismatch between the method and the empirical evidence. The paper repeatedly emphasizes a two-level MoE design. However, the default setting in the main table uses N=1. Under this setting, the second level does not actually form a mixture. The main experiments don’t truly validate the role of MoE.
2. The proof of Theorem 4.1 is not fully convincing. The appendix appears to establish the connection by imposing strong constraints on the instruction stream, so that it numerically collapses to the reconstruction value. This does lead to a unified formulation at the notation level. However, it does not seem sufficient to show strict algorithmic equivalence to the original QINCo formulation.
3. Traditional methods are evaluated with FAISS, while the neural baselines use a different implementation stack.
UNQ does not seem to be consistently rerun under the same protocol.
The paper explicitly discusses QINCo2, but does not include a direct comparison with it.
The efficiency evaluation is also under-specified. The paper reports latency numbers, but omits key details such as batch size, warm-up, synchronization, and whether memory overhead is included.
4. The presentation is somewhat imprecise in terms of dimensions and notation. In the first-layer definition, the expert component is specified in R^(D_e ), whereas the instruction vector is later written as lying in R^Din the dual-stream formulation. The instruction stream is then updated through simple addition. However, the main method description does not clearly explain how this dimensional compatibility is ensured.
5. There appears to be a minor typo on the first page. Specifically, there seems to be an extra parenthesis in the second-to-last line of the left column.

---

> ### Author Rebuttal · Authors · 2026-03-31
>
> Dear Reviewer Xp8D,
>
> We sincerely thank the reviewer for the detailed review. We appreciate the recognition of our dual-stream design and the practical value of our parallel decoding. Below we address the specific concerns raised.
>
> **Q1: The setting of N and the role of MoE.**
>
> The dual-stream design is enabled by the first-level MoE (with gate and index reuse), which provides an implicit routing mechanism without introducing additional overhead. This component is essential for decoupling the instruction stream from the quantization stream. In the main table, we set N=1 for the second-level MoE for two reasons: (1) Consistency across datasets: the optimal number of experts varies across datasets, and using N=1 ensures a controlled comparison; (2) Complexity control: when N=1, our method introduces no additional overhead compared to QINCo, isolating the benefit of the dual-stream architecture enabled by the first-level MoE.
>
> Regarding the role of the second-level MoE, we have included an analysis of N in Table 3 and Figure 4. For instance, on BigANN, the best reconstruction performance is achieved at N=4 and L=4, demonstrating the benefit of multiple experts.
>
> Following the suggestion, we have conducted additional experiments with different N under matched settings and included a full accuracy–latency Pareto frontier in the revision (see Fig.6 in https://anonymous.4open.science/r/Rebuttal-to-RQMoE). The results confirm that the dual-stream mechanism provides the primary structural improvement, while increasing N further enhances local modeling capacity for complex data manifolds. Overall, this indicates that the dual-stream design is the core contribution, and the MoE mechanism serves as an orthogonal and extensible enhancement.
>
> **Q2: Clarification on Theorem 4.1 and relation to QINCo.**
>
> We wish to clarify that Theorem 4.1 provides a constrained reduction that recovers the functional form of QINCo under specific conditions. Specifically, when N=1 and the instruction stream is constrained to coincide with the reconstruction value at each step, the formulation of RQ-MoE reduces to that of QINCo. This result serves to demonstrate that QINCo is a special case within our framework under these constraints, rather than implying strict algorithmic equivalence in general settings. From this perspective, RQ-MoE can be understood as a decoupled generalization that separates the instruction and reconstruction streams, while QINCo corresponds to the fully coupled case. We will refine the manuscript to make this relationship and its assumptions more explicit.
>
> **Q3: Experimental fairness and completeness.**
>
> Regarding QINCo2, we have conducted additional experiments to provide a direct comparison. We integrated beam search into both methods to ensure an across-the-board comparison. The MSE results (8 bytes, 500K training subset, N*L=16) are summarized below:
>
> |Method|Deep1M|BigANN1M|FB-ssnpp1M|Contriever1M|
> |-|-|-|-|-|
> QINCo2 w/o beam search|0.15|1.40|9.01|1.60|
> RQ-MoE w/o beam search|0.14|1.29|8.85|1.52|
> QINCo2 w/ beam search|0.12|1.13|8.56|1.44|
> RQ-MoE w/ beam search|0.12|1.10|8.33|1.39|
>
> These results show that RQ-MoE consistently achieves lower reconstruction error than QINCo2 under matched settings, and benefits similarly from beam search.
>
> For the UNQ evaluation, we strictly follow the standard evaluation protocol established in the QINCo paper to ensure consistency across neural quantizers. While UNQ requires substantial tuning during training, this does not affect the fairness of our inference-time comparisons. Specifically, for latency evaluation, experiments are conducted on 4×NVIDIA RTX 3090 GPUs, using data parallelism across devices. We use a batch size of 4096, perform 10 warm-up iterations followed by 100 timed runs, and apply explicit CUDA synchronization (e.g., torch.cuda.synchronize) to ensure accurate and reproducible measurements.
>
> **Q4: Dimensional consistency and notations.**
>
> There is a typo in Section 4.2: the instruction vector should lie in $R^{D_e}$. In our implementation, the instruction stream and expert codebooks reside in $R^{D_e}$, while the base and transformed codebooks reside in $R^{D}$ to remain consistent with the input space. Dimensional compatibility is maintained via linear projections where necessary. We will correct these typos and clarify the dimensional mappings throughout the paper. The extra parenthesis on the first page will also be removed.

---

> > ### Author Rebuttal · Reviewer_Xp8D · 2026-04-04
> >
> > My concerns have been adequately addressed.

---

> > > ### Author Response · Authors · 2026-04-06
> > >
> > > Dear Reviewer Xp8D,
> > >
> > > We sincerely thank you for your time and for acknowledging that our rebuttal has fully addressed your concerns. We truly appreciate your thoughtful feedback, which has helped us significantly improve the clarity and quality of our work.
> > >
> > > If you find that the clarifications and additional experimental results have strengthened your assessment of the work, we would be grateful if you could consider updating your score accordingly.
> > >
> > > Thank you again for your professional support and for the constructive review process.

---

### Official Review · Reviewer_VPbt · 2026-03-16

**Soundness:** 4
**Presentation:** 4
**Significance:** 3
**Originality:** 4
**Overall Recommendation:** 4
**Confidence:** 3

**Summary:**

This paper investigates vector quantization and overcomes the limit of dynamic quantizers with an improved residuual quantization approach. The proposed Residual Quantization via Mixture of Experts (RQ-MoE) uses a two-level MoE with dual-stream quantization for input-dependent codebook adaptation. The proposed framework is validated with state-of-the-art or on-par performance in reconstruction and retrieval and much faster decoding than prior approaches.

**Compliance With Llm Reviewing Policy:**

Affirmed.

**Final Justification:**

I have no further questions. I keep my rating.

**Key Questions For Authors:**

The paper needs some insights on MoE and discussions on the encoding efficiency.

**Limitations:**

Yes

**Strengths And Weaknesses:**

+ The proposed RQ-MoE fully explores dynamic codebook construction and decouples instruction from quantization facilitating parallel decoding.

+ The paper provides implementation codes.

+ The paper has deep discussion about vector quantization vs residual quantization, static codebooks vs dynamic codebooks, providing solid foundation for the proposed approach.

- In the proposed framework, two-level MoE mechanism is used for dual-stream quantization. While it is an effective approach as validate by the experiments, could the paper provide more insights on why MoE matches with solving the difficulty in residual quantization?

- While the decoding is accelerated a lot, but encoding is much more time consuming, and proposed method seems not have any advantages.

---

> ### Author Rebuttal · Authors · 2026-03-31
>
> Dear Reviewer VPbt,
>
> We sincerely thank the reviewer for the constructive feedback and for recognizing the originality and performance of our RQ-MoE framework. In response to the specific points raised in your review, we provide the following clarifications.
>
> **Q1: Insights on MoE.**
>
> Our fundamental motivation is to construct input-dependent codebooks that adapt to the underlying data geometry. Such adaptive codebooks allow each quantization step to leverage contextual signals beyond the current residual, effectively "localizing" the quantization process without directly re-observing the original input. This principle of specialized processing conditioned on input characteristics aligns naturally with the core mechanism of Mixture-of-Experts (MoE).
>
> Importantly, RQ can be viewed as a degenerate form of an MoE system, where Euclidean distance serves as a routing mechanism and the selected codeword corresponds to a top-1 expert. Building on this perspective, our method generalizes RQ through a two-level MoE design. Specifically, the first-level MoE reuses the quantization index as the routing signal and introduces an instruction stream to accumulate context across steps, thereby decoupling the quantization process from codebook adaptation. The second-level MoE treats experts as a set of basis functions over local subspaces and combines them through a gating mechanism to perform input-dependent transformations. This decomposition allows the model to first identify a suitable region and then adapt the codebook within that region, instead of relying on a single shared transformation. Compared to using a single deep MLP, this design is more stable in practice and avoids strong sequential coupling. Overall, this explains why MoE provides a natural and effective mechanism for addressing the limitations of residual quantization.
>
> **Q2: Encoding efficiency of RQ-MoE.**
>
> The primary computational cost of encoding in our method comes from transforming the full codebook at each step through multiple layers of second-level MoE modulation. To address this, we explore two practical optimization strategies. First, we adjust the MoE structure by increasing the number of experts while reducing their depth under a fixed budget, which improves efficiency via intra-step parallelism. Second, we apply a pre-selection strategy: instead of transforming the entire codebook, we first compute lightweight Euclidean distances to select the top-A candidates, and only apply the MoE transformation to this subset. Combined with beam search (beam size = 16)  during encoding on the BigANN1M dataset, setting A=16 achieves an 8× speedup (96.2$\mu$s to 11.4$\mu$s) with only marginal degradation in MSE (1.29 to 1.31).
>
> However, we emphasize the asymmetry between encoding and decoding in practical large-scale retrieval systems: encoding is typically a one-time cost per database vector, while decoding is executed repeatedly during large-scale queries. Therefore, optimizing decoding latency is often more critical for overall system performance.
>
> We will incorporate these analyses into the revised manuscript.

---

### Decision · Program_Chairs · 2026-04-30

**Decision:**

Accept (regular)

**Comment:**

This paper explores vector compression via residual quantization enhanced with a MoE architecture. The proposed RQ-MoE framework integrates a two-level MoE with dual-stream quantization, dynamically adapting codebooks to input geometry while enabling parallel decoding. Reviewers consider the problem is important and the solution is novel. Several issues have been raised, such as insights on MoE,  encoding efficiency of RQ-MoE, the setting of N and the role of MoE, clarification on Theorem 4.1,  whether parallel decoding relies on weaker conditioning,  whether the decoupled design is functionally meaningful, comparison under 10M training setting. Most reviewers were satisfied with the rebuttals and either increased their scores or acknowledged that the concerns were fully resolved. Accordingly, this paper is recommended for acceptance.